# Count Counts: Motivating Exploration in LLM Reasoning with Count-based Intrinsic Rewards

**Xuan Zhang**[1,2,3], **Ruixiao Li**[1,2], **Zhijian Zhou**[1,2,3], **Long Li**[1], **Yulei Qin**[3], **Ke Li**[3], **Xing Sun**[3], **Xiaoyu Tan**[3†], **Chao Qu**[1†], **Yuan Qi**[1†]

[1]Fudan University, [2]Shanghai Innovation Institute, [3]Tencent Youtu Lab
[†]Corresponding authors
xuanzhang24@m.fudan.edu.cn, quchao@fudan.edu.cn

## ABSTRACT

Reinforcement Learning (RL) has become a compelling way to strengthen the multi step reasoning ability of Large Language Models (LLMs). However, prevalent RL paradigms still lean on sparse outcome-based rewards and limited exploration, which often drives LLMs toward repetitive and suboptimal reasoning patterns. In this paper, we study the central question of how to design exploration for LLM reasoning and introduce MERCI (**M**otivating **E**xploration in LLM **R**easoning with **C**ount-based **I**ntrinsic Rewards), a novel RL algorithm that augments policy optimization with a principled intrinsic reward. Building on the idea of count-based exploration, MERCI leverages a lightweight Coin Flipping Network (CFN) to estimate the pseudo count and further epistemic uncertainty over reasoning trajectories, and converts them into an intrinsic reward that values novelty while preserving the learning signal from task rewards. We integrate MERCI into some advanced RL frameworks like Group Relative Policy Optimization (GRPO). Experiments on complex reasoning benchmarks demonstrate that MERCI encourages richer and more varied chains of thought, significantly improves performance over strong baselines, and helps the policy escape local routines to discover better solutions. It indicates that our targeted intrinsic motivation can make exploration reliable for language model reasoning.

## 1 INTRODUCTION

Reinforcement learning (RL) (Sutton & Barto, 2018) has become a cornerstone of advancing the multi-step reasoning capabilities of Large Language Models (LLMs), enabling them to tackle complex domains like competitive mathematics and code generation (Jaech et al., 2024; Guo et al., 2025; MAA, 2025). However, these tasks feature sparse rewards, with feedback available only after completing a lengthy reasoning chain, making exploration a critical challenge. Recent breakthroughs, such as Group Relative Policy Optimization (GRPO) (Shao et al., 2024) and Dynamic sAmpling Policy Optimization (DAPO) (Yu et al., 2025), have streamlined the training process by eliminating the need for an explicit value function. This yields local variability at the token level, but it does not produce exploration that is coherent across the length of a reasoning trajectory. To guide exploration in such frameworks, many prevalent techniques rely on entropy regularization to encourage local policy diversity. While effective, this approach is limited for complex, long-horizon tasks. We see an opportunity to design complementary strategies that provide more directed, temporally-consistent exploration signals particularly for those tasks, motivating our investigation into principled exploration strategies compatible with modern value-free RL.

The exploration-exploitation trade-off is a classic challenge in RL (Jin et al., 2018; Azar et al., 2017). Simple approaches such as $\epsilon$-greedy (Mnih et al., 2015) or Boltzmann exploration with entropy-based regularization (Mnih et al., 2016), inject undirected noise to encourage stochasticity (Osband et al., 2016a). While these *"shallow"* exploration methods visit all states theoretically, they can be exponentially inefficient in simple yet illustrative examples (Osband et al., 2016b; Kakade, 2003). In notoriously difficult exploration tasks like the video game *Montezuma's Revenge*, these methods fail

because the chance of discovering long, precise action sequences needed for reward is vanishingly small. In contrast, *"deep exploration"* strategies are both theoretically and empirically superior in such scenarios. These methods follow the principle of "optimism in the face of uncertainty" (Kearns & Singh, 2002; Brafman & Tennenholtz, 2002; O'Donoghue et al., 2017), encouraging the agent to explore regions of the state-action space where its knowledge is limited. This is often implemented by generating an **intrinsic reward** to densify the sparse signal from the environment. Canonical examples include pseudo-counts (Bellemare et al., 2016; Ostrovski et al., 2017), Bootstrapped DQN (Osband et al., 2016a), Random Network Distillation (RND) (Burda et al., 2019), the intrinsic curiosity module (ICM) (Pathak et al., 2017), and methods based on the Uncertainty Bellman Equation (UBE) (O'Donoghue et al., 2017).

Although desirable, existing methods for estimating epistemic uncertainty (Mannor et al., 2007) do not scale to modern LLMs. Deep Ensembles (Osband et al., 2016a; Lakshminarayanan et al., 2017), which train multiple independent models, are prohibitively expensive. Monte Carlo dropout (Gal & Ghahramani, 2016), though cheaper, still adds significant inference overhead. Other methods face architectural or theoretical hurdles: pseudo-count techniques (Bellemare et al., 2016; Ostrovski et al., 2017) depend on normalized probability densities and preclude efficient batching, while curiosity-driven methods (Burda et al., 2019; Pathak et al., 2017) lack theoretical guarantees on how the exploration bonus should decay. The UBE framework (O'Donoghue et al., 2017), while principled, relies on estimating local uncertainty, a notoriously difficult task often relegated to heuristics. This fundamental mismatch between classic uncertainty quantification and the scale of LLMs necessitates a novel approach.

Our work is founded on a critical insight applicable to a broad class of LLM reasoning tasks—specifically those that are self-contained, such as mathematical problem-solving, where the model operates without an external, stochastic world. **In this context of autoregressive generation, the underlying Markov Decision Process (MDP) has known and deterministic transitions.** When an LLM in a state $s$ (the token sequence generated so far) selects an action $a$ (the next token), the subsequent state $s' = (s, a)$ is determined without ambiguity. This property dramatically simplifies the Uncertainty Bellman Equation, which propagates uncertainty from two sources: the reward function estimate ($\hat{r}$) and the transition function estimate ($\hat{P}$). With known transitions, the epistemic uncertainty of $\hat{P}$ is zero. The UBE thus reduces to a simple accumulation of local reward uncertainty along a trajectory. This reframes the intractable problem of estimating Q-value uncertainty into the more manageable one of estimating local reward uncertainty. To make this tangible, we propose to proxy this uncertainty using a measure of state novelty—a practical and effective approach in sparse-reward settings. To this end, we employ the "Flipping Coins" method (Lobel et al., 2023), a computationally lightweight and theoretically grounded pseudo-counting technique that provides a scalable estimator for this purpose. We formalize this entire approach in our proposed algorithm, MERCI (**M**otivating **E**xploration in LLM **R**easoning with **C**ount-based **I**ntrinsic Rewards). Our code is publicly available at `https://github.com/dd88s87/MERCI`.

To our knowledge, this is the first work to derive and apply a deep exploration algorithm for LLM reasoning directly from a principled simplification of the UBE. By recognizing that **the LLM serves as its own perfectly known world model**, we bridge the gap between model-aware RL theory and the typically model-free application of RL to LLMs. Our method integrates this simplified UBE framework with the "Flipping Coins" pseudo-count module to generate an **intrinsic reward**. This reward, expressed as an exploration bonus, guides policy optimization algorithms like GRPO to explore novel reasoning trajectories based on a coherent, temporally-consistent signal of epistemic uncertainty. Experiments on complex reasoning benchmarks demonstrate that this approach significantly improves performance, effectively mitigating the tendency of standard algorithms to converge on repetitive and suboptimal solutions. Our main contributions are summarized as follows:

1. **A Novel Theoretical Framework for LLM Exploration.** We establish a new framework based on a key insight: the LLM's known and deterministic transition dynamics simplify the Uncertainty Bellman Equation. This renders principled, uncertainty-driven exploration tractable at scale by reducing the intractable problem of Q-value uncertainty to a manageable estimation of local reward uncertainty.

2. **A Practical and Scalable Exploration Algorithm.** We propose **MERCI**, a novel algorithm that operationalizes our theoretical framework. MERCI employs a highly scalable

counting method to translate state novelty into a potent intrinsic reward signal, designed for seamless integration with modern, value-free policy optimization methods like GRPO.

3. **State-of-the-Art Performance on Complex Reasoning.** Our extensive empirical evidence on challenging reasoning benchmarks, including MATH and SQL generation, demonstrate that MERCI beats strong baselines. Its directed exploration mechanism mitigates premature convergence and leads to the discovery of more robust and accurate solutions.

## 2 PRELIMINARIES

**Coin Flip Network (CFN)**   CFN is a computationally efficient method of count-based exploration, which estimates a state's visitation count by solving a simple regression problem. The core idea is that a state's visitation count can be estimated by leveraging the statistical properties of the Rademacher distribution (i.e., random coin flips) (Lobel et al., 2023). The method works by setting up a supervised learning problem where a neural network $f_\phi$, i.e., the CFN, is trained to predict the average of random coin-flip vectors associated with each state it encounters.

For every visit to a state $s_i$, a new random vector $y_i$ (i.e., the *coin flips*) is sampled from $\{-1, 1\}^d$. The CFN $f_\phi$ is learned by solving $\arg\min_\phi \mathbb{E}_{(s_i, y_i) \sim \mathcal{D}_{cfn}}[\mathcal{L}(s_i, y_i)]$, where $\mathcal{L}$ is the mean-square error loss function and $\mathcal{D}_{cfn}$ is a dataset of state-label pairs. Considering the fair coin-flip distribution $\mathcal{C}$ over outcomes $\{-1, 1\}$, we can flip this coin $n$ times and average the results into $z_n$. Specifically, the second moment of the sample mean $z_n$ is related to the inverse count: $\mathcal{M}_2(z_n) = \mathbb{E}[z_n^2] = \sum_i Pr(z_n = i) * i^2 = \frac{1}{n}$. $\mathbb{E}[z_n^2]$ is the variance of the sample mean of the coin-flip distribution. Furthermore, by flipping $d$ coins each time, the variance of $z_n^2$ can be reduced by a factor of $\frac{1}{d}$, which implies a reliable way for estimating the inverse count. To this end, we generate a $d$-dimensional random vector $c_i \sim \{-1, 1\}^d$ as a label $y_i$ for state $s_i$. The learning objective is described as:

$$f_\phi^*(s) = \arg\min_\phi \mathbb{E}_{(s_i, y_i) \sim \mathcal{D}_{cfn}}[\mathcal{L}(s_i, y_i)] = \arg\min_\phi \sum_{i=1}^{|\mathcal{D}_{\text{cfn}}|} \|\mathbf{c}_i - f_\phi(s_i)\|^2. \tag{1}$$

In the dataset $\mathcal{D}_{cfn}$, each occurrence of the same state will be paired with a different random vector. $f_\phi^*$ cannot learn a perfect mapping from states to labels since there are more than one (i.e., $m$) instances of the same state $s_i$. Thus, it instead minimizes $\mathcal{L}$ by outputting the mean random vector for all instances of a given state: $f_\phi^*(s) = \frac{1}{n} \sum_{i=1}^n \mathbf{c}_i$. The pseudo-count can be estimated by:

$$\frac{1}{d}\|f_\phi(s)\|^2 = \frac{1}{d}\sum_{j=1}^d \mathbb{E}\left[\left(\sum_{i=1}^n \frac{c_{ij}}{n}\right)^2\right] = \frac{1}{d}\sum_{j=1}^d \mathbb{E}\left[z_n^2\right] = \frac{1}{n}. \tag{2}$$

By training $f_\phi$ on the objective described in Equation 1, we can map states to approximate the count by: $\frac{1}{d}\|f_\phi(s)\|^2 \approx \frac{1}{\mathcal{N}(s)}$, where $\mathcal{N}(s)$ denote the counts of state $s$.

**Group Relative Policy Optimization (GRPO)**   GRPO (Shao et al., 2024) discards the value network in PPO (Schulman et al., 2017) by calculating the advantage of each reasoning step against the value of the entire completed sequence. For each question $q$ and its ground-truth answer $a$, GRPO samples a group of outputs $\{o_i\}_{i=1}^G$ from the old policy $\pi_{\theta_{\text{old}}}$ with corresponding outcome rewards $\{R_i\}_{i=1}^G$, and then computes the normalized reward in each group as the estimated advantage:

$$\hat{A}_t^i = \frac{r_i - \text{mean}\left(\{R_i\}_{i=1}^G\right)}{\text{std}\left(\{R_i\}_{i=1}^G\right)}, \quad \text{where } R_i = \begin{cases} 1.0 & \text{if is\_equivalent}(a, o_i), \\ 0.0 & \text{otherwise.} \end{cases} \tag{3}$$

Adding a KL penalty term to the clipped objective in PPO, the objective of GRPO is expressed as:

$$\mathcal{J}_{\text{GRPO}}(\theta) = \mathbb{E}_{(q,a) \sim \mathcal{D}, \{o_i\}_{i=1}^G \sim \pi_{\theta_{\text{old}}}(\cdot|q)} \left[ \frac{1}{G}\sum_{i=1}^G \frac{1}{|o_i|}\sum_{t=1}^{|o_i|} \left( \min\left(r_t^i(\theta)\hat{A}_t^i, \text{clip}(r_t^i(\theta), 1-\epsilon, 1+\epsilon)\hat{A}_t^i\right) \right. \right.$$
$$\left. \left. - \beta \, \mathbb{D}_{\text{KL}}\left[\pi_\theta \| \pi_{\text{ref}}\right] \right) \right], \quad \text{where } r_t^i(\theta) = \frac{\pi_\theta(o_{i,t}|q, o_{i,<t})}{\pi_{\theta_{\text{old}}}(o_{i,t}|q, o_{i,<t})}. \tag{4}$$

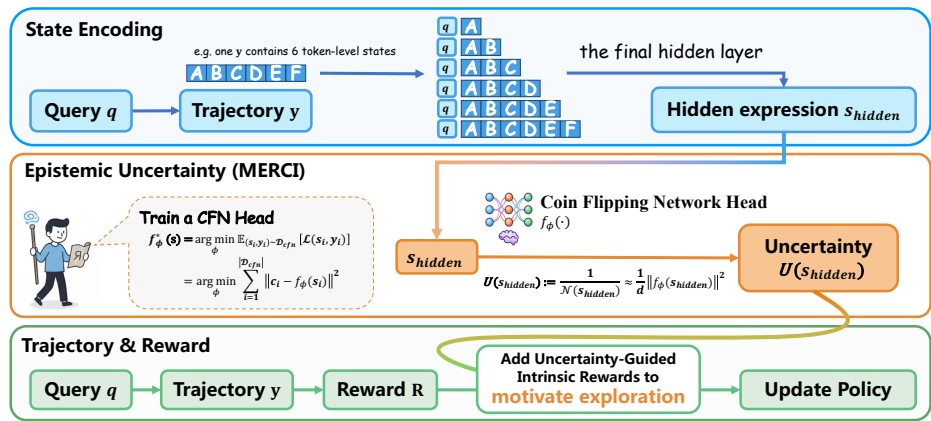

Figure 1: Overview of the MERCI framework. Two separate networks are used: a policy network $\pi_\theta$ trained with RL, and a CFN network that provides an intrinsic reward. The CFN network, initialized from the same SFT checkpoint $\pi_0$, estimates state novelty to guide the exploration of $\pi_\theta$.

**Decouple Clip and Dynamic sAmpling Policy Optimization (DAPO)**  Building on GRPO, DAPO (Yu et al., 2025) removes the KL penalty, introduces a clip-higher strategy and dynamic sampling, applies a token-level policy gradient loss, and adopts overlong reward shaping.

## 3 METHODOLOGY

In this section, we first establish the theoretical foundation for our approach by simplifying the Uncertainty Bellman Equation for the specific case of LLMs, and then introduce the full details of our novel algorithm, **MERCI**.

### 3.1 THE UNCERTAINTY BELLMAN EQUATION WITH KNOWN TRANSITIONS

The Uncertainty Bellman Equation (UBE) provides a principled mechanism for propagating epistemic uncertainty—quantified as the variance of the posterior distribution over Q-values—through time (O'Donoghue et al., 2017). For clarity, we will use the terms "uncertainty" and "variance" interchangeably throughout this section. Our core theoretical contribution stems from a key insight: **the Markov Decision Process (MDP) underlying LLM reasoning has a known and deterministic transition function,** $P$. This property dramatically simplifies the general form of the UBE, leading to a more direct and tractable equation for uncertainty propagation.

Formally, we consider a *finite* horizon, finite state and action space MDP, with horizon length $H \in \mathbb{N}$, state space $\mathcal{S}$, action space $\mathcal{A}$ and rewards at time period $h$ denoted by $r^h \in \mathbb{R}$. A policy $\pi = (\pi^1, \ldots, \pi^H)$ is a sequence of functions where each $\pi^h : \mathcal{S} \times \mathcal{A} \to \mathbb{R}_+$ is a mapping from state-action pair to the probability of taking that action at that state, i.e., $\pi_{sa}^h$ is the probability of taking action $a$ at state $s$ at time-step $h$ and $\sum_a \pi_{sa}^h = 1$ for all $s \in \mathcal{S}$. At each time-step $h$ the agent receives a state $s^h$ and a reward $r^h$ and selects an action $a^h$ from the policy $\pi^h$, and the agent moves to the next state $s^{h+1}$, which is sampled with probability $P_{s'sa}^h$. The Q-value, at time step $h$ of a particular state under policy $\pi$ is the expected total return from taking that action at that state and following $\pi$ thereafter, i.e., $Q^{\pi,h}(s,a) = \mathbf{E}\left[\sum_{t=h}^H r^t \mid s^t = s, a^t = a, \pi\right]$.

We adopt a Bayesian perspective as that in (O'Donoghue et al., 2017). We assume a prior over the mean reward function, $r(s)$, and collect a history of interactions $\mathcal{F}_t$ (states, actions, and rewards up to episode $t$) generated by a policy $\pi$. This history is used to form a posterior distribution over the mean rewards, which we denote $\phi_{r|\mathcal{F}_t}$. If we draw a reward function estimate $\hat{r} \sim \phi_{r|\mathcal{F}_t}$, the corresponding Q-function estimate, $\hat{Q}^\pi$, must satisfy the posterior Bellman equation for that sample:

$$\hat{Q}^{\pi,h}(s,a) = \hat{r}^h(s) + \sum_{s',a'} \pi_{s',a'}^h P_{s'sa}^h [\hat{Q}^{\pi,h+1}(s',a')], \tag{5}$$

for all timesteps $h = 1, \ldots, H$, with $\hat{Q}^{\pi, H+1} = 0$.

Since the transition function $P$ for an LLM is a known delta function (i.e., for a given state $s$ and action $a$, the next state $s' = (s, a)$), we have $P$ rather than its posterior $\hat{P}$ in equation 5. This leads to a recursive equation for the variance of the Q-value posterior, as stated in the following proposition. In the following discussions, we may use the word uncertainty and variance (w.r.t. the posterios distribution) interchangeably. we denote $\mathbb{V}_t x$ as the variance of random variable $x$ conditioned on the history $\mathcal{F}_t$, which is $\mathbb{E}\left((x - \mathbb{E}(x|\mathcal{F}_t))^2 \Big| \mathcal{F}_t\right)$.

**Proposition 1 (Uncertainty Bellman Equation for Known Transitions)** *Let* $U^h(s, a) \triangleq \mathbb{V}_t[\hat{Q}^{\pi, h}(s, a)]$ *be the posterior variance of the Q-value at step* $h$, *conditioned on the history* $\mathcal{F}_t$. *Given a known and deterministic transition function, this uncertainty propagates according to the following Bellman equation:*

$$U^h(s, a) \leq \mathbb{V}_t[\hat{r}^h(s)] + \sum_{s', a'} \pi^h_{s', a'} P^h_{s'sa} U^{h+1}(s', a'),$$

*where* $s'$ *is the unique next state reached from* $(s, a)$, *and* $U^{H+1}(\cdot) = 0$.

The proof follows from the analysis in O'Donoghue et al. (2017) by applying the law of total variance to equation 5. This result provides a powerful recursive formula: **the uncertainty of a state-action pair is bounded by the immediate reward uncertainty plus the expected uncertainty of the unique subsequent state**, where the expectation is over the policy's next actions. This reframes the complex problem of estimating Q-value variance into the more manageable task of estimating the local reward uncertainty, $\mathbb{V}_t[\hat{r}^h(s)]$. The resulting Q-value variance, $U^h(s, a)$, can be used to define an exploration bonus inspired by Upper Confidence Bound (UCB) algorithms (Lattimore & Szepesvári, 2020). Specifically, the policy can be encouraged to explore by modifying the optimization objective to $Q^{\pi, h}(s, a) + \alpha \sqrt{U^h(s, a)}$, where $\alpha$ is a hyperparameter balancing exploitation and exploration. This approach is backed by strong theoretical guarantees for achieving low regret (Auer et al., 2008; Jin et al., 2018).

From standard concentration inequalities, we know that the uncertainty over a mean reward estimate is inversely proportional to the number of times that state has been visited, i.e., $\mathbb{V}_t[\hat{r}^h(s)] \propto 1/\mathcal{N}(s)$. However, in the high-dimensional state space of language, exact state visitations are exceedingly rare. This necessitates a method to generalize counting to unseen but similar states. In the following section, we describe how we use a scalable pseudo-count mechanism to estimate this local uncertainty.

## 3.2 ESTIMATE VARIANCE OF REWARD VIA CFN

Standard policy optimization driven by sparse, outcome-based rewards (e.g., GRPO) can lead to premature convergence on suboptimal solutions. MERCI addresses it via a dedicated mechanism for principled exploration. The framework is illustrated in Figure 1.

Our framework employs two distinct Large Language Models operating in parallel:

1. **The Policy Network ($\pi_\theta$):** This is the agent that generates reasoning trajectories. It is initialized from a supervised fine-tuned (SFT) checkpoint, $\pi_0$, and its parameters $\theta$ are exclusively updated by the policy optimization algorithm (e.g., GRPO).

2. **The CFN Network:** This network's sole purpose is to estimate epistemic uncertainty. It is a separate instance of the LLM, also initialized from the same checkpoint $\pi_0$. A lightweight MLP, which we call the CFN head ($f_\phi$), is attached to its final hidden layer. CFN network is updated together via a supervised regression objective (detailed in Section 2).

The training process integrates these two networks as follows. During a training step, a reasoning trajectory $\tau$ is first generated by the current policy network $\pi_\theta$. In the sequential decision-making process, we define the state at each step as the contextual hidden representation $s_{\text{hidden}}$ output by the LLM backbone at that token position, which inherently captures the entire prefix of the generated sequence. The state $s_{\text{hidden}}$ is then processed by the CFN head $f_\phi(s)$ to estimate the variance of the reward, computed by $\mathbb{V}[\hat{r}(s)] = \frac{1}{d}\|f_\phi(s)\|^2$.

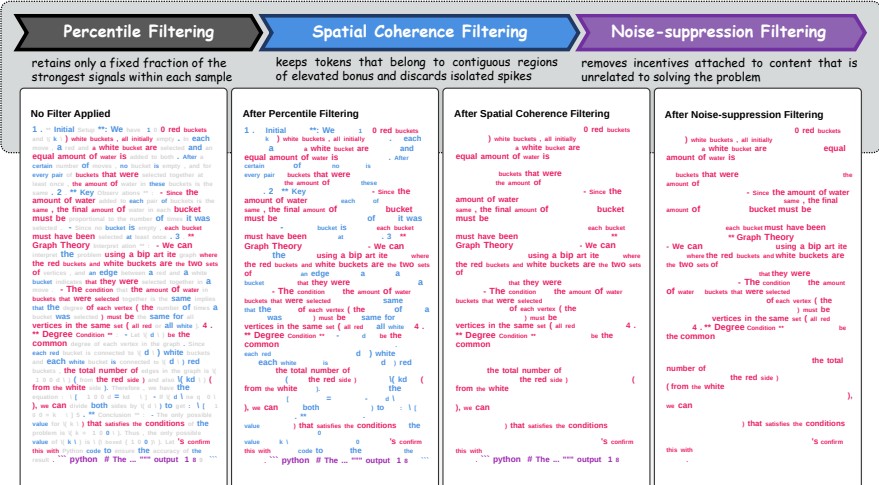

Figure 2: The entire pipeline of bonus filtering. Step 1: We rank all tokens within a response by their associated bonus values and retain only those falling within a predefined top percentile (e.g., the top 50% in this figure). Step 2: We only preserve clusters of adjacent tokens that consistently exhibit elevated bonuses (e.g., 3 consecutive tokens in this figure). Step 3: For example, in a math reasoning task without external tools, any Python code potentially generated during LLM rollouts is semantically irrelevant and noisy, so we exclude them from the overall bonus calculation.

## 3.3 ADVANTAGE ESTIMATION

**Calculating the Intrinsic Reward from Cumulative Uncertainty**    A critical detail of our method, derived directly from Proposition 1, is the precise calculation of the exploration bonus. The correct approach to determine the uncertainty of a trajectory's value is to first **sum the local reward variances** at each step (we use the monte carlo estimation here), and only then take the square root of the total sum. This resulting value represents the standard deviation of the cumulative Q-value posterior and serves as our intrinsic reward.

This stands in stark contrast to a common but theoretically flawed heuristic in many RL exploration algorithms. Those methods often compute a per-step bonus proportional to the local *standard deviation* and apply a standard RL algorithm to the modified, "bonused" rewards. As demonstrated by O'Donoghue et al. (2017), this latter approach—which is equivalent to summing standard deviations—leads to a significant overestimation of uncertainty over long horizons. This miscalculation can cause the agent to become overly optimistic, leading to prolonged and inefficient exploration of paths that are long but not necessarily promising. To illustrate the difference, consider a trajectory of horizon $H$ where the local reward variance at each step is $\sigma^2 = 1$. **Correct Bonus (MERCI):** The cumulative variance is $\sum_{h=1}^{H} 1 = H$. The bonus, or standard deviation, is correctly calculated as $\sqrt{H}$. **Heuristic Bonus:** The per-step bonus is $\sqrt{1} = 1$. Summing these bonuses results in an overestimated total bonus of $\sum_{h=1}^{H} 1 = H$. **MERCI** adheres strictly to the former, theoretically-grounded calculation, ensuring the exploration signal accurately reflects the true cumulative epistemic uncertainty. Indeed, we compare this two calculation in our ablation study G.2.4.

**Budget-Aware Exploration Bonus Control**    The non-sparse exploration bonus introduces its own considerable instabilities when becoming indiscriminately dense, which would invite LLMs seeking through aimless exploration. So, we enforce budgeted exploration, which reduces gradient variance and in turn stabilizes optimization and lowers noise in final answers. Concretely, three filtering stages are applied, shaping where and how the bonus can act. (1) **Percentile filtering** retains only a fixed fraction of the strongest signals within each sample, which tracks the gradual decline in bonus magnitude over training without manual retuning. (2) **Spatial coherence filtering** keeps tokens that belong to contiguous regions of elevated bonus and discards isolated spikes even when numerically large, thereby yielding steadier updates. (3) **Noise-suppression filtering** removes incentives attached to content that is unrelated to solving the problem, such as meaningless repetition, gratuitous

code blocks, or rare characters generated solely to chase the bonus. Together these stages allocate a controlled exploration budget that preserves useful exploration while safeguarding the primary reward signal. The overall pipeline of bonus control is illustrated in Figure 2.

**Advantage Normalization and Bonus Integration**     After bonus filtering, the normalized bonus $\mathcal{B}$ is computed by first averaging the squared CFN outputs across all retained tokens and then applying square-root compression:

$$\mathcal{B} = \sqrt{\frac{1}{l} \sum_{i \in \mathbb{I}} \left( \frac{1}{d} \| f_\phi(s^i_{hidden}) \|^2 \right)}, \tag{6}$$

where $l$ is the length of a trajectory, $d$ is the dimension of CFN's outputs, and $\mathbb{I}$ is the set of retained tokens' indices.

To ensure comparability across trajectories sampled under the same prompt, we standardize trajectory-aggregated bonuses within each group of size $G$ and truncate negative values, preserving only positive exploratory incentives:

$$\hat{A}^i_{\text{exploration}} = \max \left( 0, \frac{\mathcal{B}_i - \mu}{\sigma} \right), \text{where} \ \ \mu = \frac{1}{G} \sum_{j=1}^{G} \mathcal{B}_j, \ \sigma = \sqrt{\frac{1}{G} \sum_{j=1}^{G} (\mathcal{B}_j - \mu)^2}. \tag{7}$$

To prevent the bonus from overpowering outcome-based rewards, we scale the standardized intrinsic bonus term by an exploration coefficient $\gamma$, and add it to the base advantage $\hat{A}^i_{\text{old}}$. For trajectories whose base advantage is negative, we cap the augmented advantage with a clipping factor $\alpha \in (0, 1)$ to prevent the intrinsic term from overwhelming the outcome signal:

$$\hat{A}^i_{\text{new}} = \begin{cases} \min \left( \hat{A}^i_{\text{old}} + \gamma \hat{A}^i_{\text{exploration}}, (1 + \alpha)\hat{A}^i_{\text{old}} \right), & \text{if } \hat{A}^i_{\text{old}} \geq 0; \\ \min \left( \hat{A}^i_{\text{old}} + \gamma \hat{A}^i_{\text{exploration}}, (1 - \alpha)\hat{A}^i_{\text{old}} \right), & \text{if } \hat{A}^i_{\text{old}} < 0. \end{cases} \tag{8}$$

We give an algorithmic description in Algorithm 1 in Appendix C.

## 4 RELATED WORK

### 4.1 REINFORCEMENT LEARNING FOR LLM REASONING

Reinforcement learning (RL) (Sutton & Barto, 2018), particularly Reinforcement Learning with Verifiable Rewards (RLVR), has been widely used to improve the reasoning abilities of large language models (LLMs). PPO is a foundational policy gradient method, which ensures stable policy updates via clipped objectives, proving effective in reasoning tasks (Schulman et al., 2017). It treats token positions in reasoning trajectories of LLM as distinct states for advantage estimation, but this approach comes at the cost of computational overhead from its joint policy-value optimization. Starting from PPO, recent efforts have developed some efficient and advanced frameworks such as GRPO (Shao et al., 2024). By evaluating and normalizing rewards across a group of entire generated sequences, GRPO provides a more robust and efficient method for advantage estimation. This method of relative, sequence-level comparison sidesteps the complexities of token-level advantage estimation, proving far more effective for multi-step reasoning. The success of this holistic approach is highlighted by its adoption and extension in subsequent research, such as DAPO (Yu et al., 2025), VAPO (Yue et al., 2025) and Dr. GRPO (Liu et al., 2025). However, even advanced RL methods for LLMs face a critical bottleneck: their dependence on external static and sparse reward structures limits effective exploration. To overcome this, we integrate count-based intrinsic motivation into GRPO-like frameworks, incentivizing the model to explore more novel and diverse reasoning trajectories guided by epistemic uncertainty during training.

### 4.2 EXPLORATION IN REINFORCEMENT LEARNING

Effective exploration in RL is critical for navigating the fundamental dilemma between exploiting known rewards and exploring uncertain options to discover better policies. Some traditional

exploration methods like RND (Burda et al., 2019), ICM (Pathak et al., 2017), and Count-Based Exploration (Ostrovski et al., 2017; Tang et al., 2017a), encourage agents to explore novel or under-visited states via intrinsic rewards. However, their application to LLMs faces significant challenges: the dynamic response length and vast action space. Most approaches for LLMs rely on undirected exploration, such as simply encouraging exploration from an entropy perspective (Wen et al., 2024; Wang et al., 2025; Cheng et al., 2025). These heuristic approaches often lack solid theoretical foundation to guide policy models to identify which states warrant greater exploration, leading to suboptimal policies. To address these limitations, recent work has developed active exploration strategies to estimate uncertainty from historical data and plan optimistically (Zhang et al., 2024; Bai et al., 2025; Cen et al.; Chen et al., 2025; Gao et al., 2025; Zhang et al., 2025; Dai et al., 2025). However, curiosity-driven methods (Bougie & Watanabe, 2025; Gao et al., 2025; Dai et al., 2025) lack theoretical guarantees on how the exploration bonus should decay, and the classical density-based methods for calculating pseudo-counts (Ostrovski et al., 2017; Bai et al., 2021) are resource-intensive, time-consuming, and hard to fulfill. Some methods (Tang et al., 2017b; Rashid et al., 2019; Lobel et al., 2023) instead explored alternatives to eliminate the usage of density models. In our work, we formally show that the deterministic nature of LLM transitions simplifies the general Uncertainty Bellman Equation to a tractable form, providing the principled justification for how to aggregate the local pseudo-counts into a sum-of-variance trajectory bonus, thereby distinguishing our method from purely heuristic exploration techniques. We take CFN (Lobel et al., 2023) as our theoretical foundation for estimating the pseudo-count, introducing a simple supervised learning objective to estimate a visitation count and further integrates intrinsic motivation.

## 5    EXPERIMENTS

To validate our hypothesis that encouraging novelty via MERCI promotes the policy's ability to discover more optimal solutions, we conduct a comprehensive set of experiments on two types of benchmarks: mathematical reasoning and SQL generation, and further provide in-depth analyses.

### 5.1    EXPERIMENTAL SETUP

**Mathematical Reasoning**    Our backbone model is Qwen2.5-Math-7B (Yang et al., 2024). Our training dataset is sourced from DAPO-17K (Yu et al., 2025), and we evaluate models on a diverse set of challenging mathematical reasoning benchmarks, including AIME2024/2025 (MAA, 2025), MATH500 (Hendrycks et al., 2021), OlympiadBench (He et al., 2024), College Math (Tang et al., 2024), and Minerva (Lewkowycz et al., 2022).

**SQL Generation**    Our experiments are conducted on Llama-3.1-8B-Instruct (Grattafiori et al., 2024). We trained on the Bird training set (Li et al., 2023) and evaluated performance on the Bird and Spider test sets (Yu et al., 2019).

**Baselines and Configurations**    We conduct RL training experiments on both vanilla GRPO and DAPO using the veRL framework (Sheng et al., 2025). We additionally introduce two algorithms designed to encourage exploration as baselines: one uses entropy-based advantage shaping(Cheng et al., 2025), and the other incorporates intrinsic rewards via RND training (Gao et al., 2025). In our experimental results, we refer to them as Entropy Adv. and iMentor, respectively. For the implementation of CFN, we set the dimensionality $d$, which can be intuitively interpreted as *how many times we have flipped a coin*, to 20. Detailed hyperparameters are presented in Appendix E.

### 5.2    MAIN RESULTS

#### 5.2.1    COIN FLIP NETWORK

To evaluate the effectiveness of the CFN, i.e., our exploration model, and enhance exploration efficiency during RL training, we first generate responses from the backbone model on the training dataset and use these responses to perform a preliminary training of the CFN. This process enables it to develop a basic understanding of which states are likely to occur more rarely.

Table 1: Performance on mathematical reasoning benchmarks with pass@$k$ and mean@$k$. The highlighted color represents the best within RL models, while underlined represents the second best.

(a) pass@$k$ results

| | AIME25 pass@256 | AIME24 pass@256 | Minerva pass@16 | MATH500 pass@16 | OlympiadBench pass@16 | College pass@8 | Avg. |
|---|---|---|---|---|---|---|---|
| *Qwen2.5-Math* | 53.3 | 70.0 | 50.4 | 88.6 | 56.7 | 44.2 | 60.5 |
| + GRPO | 53.3 | 76.7 | 64.0 | 91.8 | 59.7 | 49.2 | 65.8 |
| + GRPO w/ Entropy Adv. | 56.7 | 76.7 | 62.5 | 91.2 | 59.4 | 48.9 | 65.9 |
| + GRPO w/ iMentor | 60.0 | 76.7 | 61.4 | 90.4 | 60.4 | 49.3 | 66.4 |
| + GRPO w/ MERCI (ours) | 60.0 ↑ | 80.0 ↑ | 63.2 | 91.4 | 60.9 ↑ | 48.9 | 67.4 ↑ |
| + DAPO | 56.7 | 76.7 | 66.9 | 92.0 | 60.9 | 48.3 | 66.9 |
| + DAPO w/ Entropy Adv. | 60.0 | 83.3 | 66.5 | 91.4 | 57.6 | 48.5 | 67.9 |
| + DAPO w/ iMentor | 56.7 | 76.7 | 68.0 | 92.0 | 60.0 | 50.1 | 67.3 |
| + DAPO w/ MERCI (ours) | 60.0 ↑ | 83.3 ↑ | 66.5 | 91.8 | 62.1 ↑ | 50.2 ↑ | 69.0 ↑ |

(b) mean@$k$ results

| | AIME25 mean@256 | AIME24 mean@256 | Minerva mean@16 | MATH500 mean@16 | OlympiadBench mean@16 | College mean@8 | Avg. |
|---|---|---|---|---|---|---|---|
| *Qwen2.5-Math* | 4.4 | 10.7 | 16.9 | 47.5 | 64.6 | 22.1 | 20.3 |
| + GRPO | 11.2 | 28.7 | 41.8 | 79.0 | 40.3 | 42.0 | 40.5 |
| + GRPO w/ Entropy Adv. | 12.1 | 28.9 | 42.0 | 81.0 | 40.6 | 42.6 | 41.2 |
| + GRPO w/ iMentor | 11.9 | 29.0 | 42.2 | 78.9 | 40.7 | 42.4 | 40.9 |
| + GRPO w/ MERCI (ours) | 13.4 ↑ | 29.6 ↑ | 44.1 ↑ | 80.7 ↑ | 42.6 ↑ | 42.9 ↑ | 42.2 ↑ |
| + DAPO | 16.5 | 31.9 | 41.0 | 81.5 | 41.4 | 41.0 | 42.2 |
| + DAPO w/ Entropy Adv. | 17.2 | 33.3 | 44.5 | 80.9 | 41.4 | 41.6 | 43.2 |
| + DAPO w/ iMentor | 17.4 | 32.0 | 46.7 | 82.3 | 42.8 | 43.3 | 44.1 |
| + DAPO w/ MERCI (ours) | 18.4 ↑ | 35.2 ↑ | 44.8 ↑ | 82.4 ↑ | 44.3 ↑ | 44.2 ↑ | 44.9 ↑ |

For this pretrained CFN, we conduct two evaluations: (1) Within a single response, we visualize the estimated uncertainty assigned by the CFN to each token position; (2) For all collected responses, we apply the method described in Section 3.3 to select the top $30\%$ of tokens with the highest bonus in each response, filter them accordingly, and then perform statistical analysis on the retained token sequences. The results of Experiment (1) and (2) are presented in Figure 3, 4, 5, 6 and Figure 7 in Appendix G, respectively. We can observe that token sequences assigned *higher uncertainty* by the CFN predominantly correspond to *novel reasoning paths*, Python code along with its outputs, or specialized mathematical terminologies. This observation aligns with our hypothesis that more novel token positions tend to induce higher epistemic uncertainty and are therefore assigned higher values by our CFN.

In addition, the CFN exhibits three further important findings: (1) when directly applied to estimate the uncertainty of responses in the SQL Generation task, the CFN trained on mathematical reasoning tasks produces estimates that align well with our intuition and analysis, which indicates the generalization ability of CFN; (2) for reasoning trajectories that are linguistically close but not identical, the CFN successfully captures their semantic similarity, yielding correspondingly similar uncertainty estimates; (3) our CFN bonus provides a non-redundant signal and effectively measures the policy's epistemic uncertainty (i.e., lack of knowledge). The detailed results are shown in Figure 8, 9, 10 and Figure 11 in Appendix G.1.

### 5.2.2 RL TRAINING

The CFN in the RL phase is initialized using the pretrained CFN and is then co-trained with the policy model during RL training.

Our primary results for RL training are summarized in Table 1 and Table 2. As shown in Table 1, MERCI delivers consistent gains over both vanilla GRPO and DAPO across mathematical reasoning benchmarks when measured by pass@$k$ and mean@$k$. Gains are most pronounced on the AIME suites, which is the most challenging, and remain robust on the other datasets. Consistently higher mean@$k$ suggest better overall sample quality with uniform and stable gains. In addition, MERCI also yields improvements in pass@$k$, pointing to enhanced exploration and calibration rather than

Table 2: Performance on SQL generation benchmarks with greedy sampling and pass@*k*.

| Model | Bird (in domain) | | | Spider (out of domain) | | |
|---|---|---|---|---|---|---|
| | Greedy | *Pass@8* | *Pass@16* | Greedy | *Pass@8* | *Pass@16* |
| *Llama-3.1-8B-Instruct* | 42.4 | 68.5 | 75.1 | 69.0 | 91.0 | 94.6 |
| + GRPO | 60.7 | 72.2 | 74.6 | 74.7 | 81.0 | 82.9 |
| + GRPO w/ Entropy Adv. | 60.8 | 72.1 | 73.9 | 74.7 | 83.2 | 84.5 |
| + GRPO w/ iMentor | 62.8 | 72.3 | 74.2 | 75.0 | 84.1 | 85.2 |
| + GRPO w/ MERCI (ours) | 63.0 ↑ | 72.8 ↑ | 74.9 ↑ | 78.0 ↑ | 84.1 | 85.6 ↑ |
| + DAPO | 63.2 | 73.9 | 75.9 | 76.8 | 86.1 | 87.2 |
| + DAPO w/ Entropy Adv. | 62.3 | 73.2 | 75.9 | 77.5 | 86.1 | 87.6 |
| + DAPO w/ iMentor | 62.7 | 73.9 | 76.1 | 77.2 | 86.4 | 88.2 |
| + DAPO w/ MERCI (ours) | 64.1 ↑ | 73.6 | 76.1 | 77.3 | 86.9 ↑ | 88.5 ↑ |

narrow best-case gains. As shown in Table 2, SQL generation results on Bird and Spider also mirror the earlier findings. Especially, MERCI yields larger out-of-domain gains, i.e., the Spider test set. It indicates that MERCI effectively pushes LLMs to use general SQL patterns that transfer better to different schemas. Additionally, the cross-domain experiments in Appendix G.2.1 indicate that our MERCI play an important role in improving out-of-domain robustness, even when the underlying training data is highly domain-specialized.As evidenced by the training dynamics in Figure 13 in Appendix G and our case study in Appendix G.2.6, we further observe that MERCI enhances exploration and calibration by densifying multiple valid reasoning trajectories while discouraging gratuitous chain elongation. It concentrates probability mass on more diverse yet more reliable good solutions that are expressed in shorter, more focused traces, raising the floor of candidate quality. Besides, our case study also indicates an increased proportion of steps devoted to higher-level reasoning abilities. This shift from length-based search to concise, well-calibrated reasoning improves sample efficiency and reduces error correlation. It learns to prune task-irrelevant branches and concentrate computation on promising hypotheses, yielding more intelligent and efficient exploration.

## 5.3 ABLATION STUDIES AND SCALING EXPERIMENTS

We conducted these experiments on the mathematical reasoning task and vanilla GRPO. The detailed experimental results are presented in Appendix G.2.2 and Appendix G.2.4. From these results, we first confirm that crucial components, including bonus filtering and our normalized trajectory-aggregated uncertainty estimation, are fundamental to the method's success. Furthermore, the results reveal MERCI's superior exploratory efficiency: our algorithm not only identifies good solutions efficiently and yields strong pass@*k* performance, but also demonstrates remarkable stability over the long term in scaling experiments. Finally, sensitivity analysis on the key hyperparameter choices, e.g., the $\gamma$ cosine schedule and Top-$p\%$ in bonus filtering, are included in Appendix G.2.5.

## 6 CONCLUSION

In this study we introduced MERCI, a principled exploration strategy for LLM reasoning that harnesses the deterministic transitions of language trajectories. By reframing the Uncertainty Bellman Equation under known transitions we replaced expensive Q variance estimation with a tractable count based proxy for reward uncertainty. The result is an intrinsic signal that guides Group Relative Policy Optimization and its variants toward diverse and coherent reasoning paths. Experiments on challenging mathematics and SQL benchmarks reveal consistent gains in pass rates and in mean score, verifying that our method steers policies away from shallow entropy driven randomness and toward productive inquiry. The Coin Flip Network delivers this benefit with minimal compute overhead and can be trained in parallel with the policy model, which makes the approach attractive for large scale systems. Experiments on mathematical reasoning and SQL generation show stable training dynamics, diverse reasoning paths, accurate solutions, and robust outcomes at scale.

## 7 ACKNOWLEDGEMENT

This work was supported by the National Natural Science Foundation of China (Grant No. 82394432, and 92249302), and the Shanghai Municipal Science and Technology Major Project (Grant No. 2023SHZDZX02).

## 8 ETHICS STATEMENT

This work adheres to the ICLR Code of Ethics. In this study, no human subjects or animal experimentation was involved. All datasets used were sourced in compliance with relevant usage guidelines, ensuring no violation of privacy. We have taken care to avoid any biases or discriminatory outcomes in our research process. No personally identifiable information was used, and no experiments were conducted that could raise privacy or security concerns. We are committed to maintaining transparency and integrity throughout the research process.

## 9 REPRODUCIBILITY STATEMENT

We have made every effort to ensure that the results presented in this paper are reproducible. All models and datasets used in our work are publicly available, and the code is openly available at: `https://github.com/dd88s87/MERCI`. The experimental setup, including training steps, model configurations, and hardware details, is described in detail in the appendix. We have also provided a full description to assist others in reproducing our experiments.

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

## A   THE USE OF LARGE LANGUAGE MODELS

We utilize an LLM to assist with paper editing and correcting grammatical errors.

## B   ENTROPY REGULARIZATION AS A GENERALIZED $\epsilon$-GREEDY EXPLORATION

We provide a mathematical derivation showing that entropy regularization corresponds to a softmax exploration strategy. This can be interpreted as a generalized form of $\epsilon$-greedy exploration that intelligently allocates exploration probability based on the relative quality of suboptimal actions.

**Entropy-regularized policy improvement.**   Given a state $s$ and advantage estimates $A(s, a)$ for actions $a \in \mathcal{A}$, consider the entropy-regularized optimization:

$$\pi^\star = \arg \max_{\pi(\cdot|s)} \sum_a \pi(a|s) \, A(s, a) + \beta H(\pi(\cdot|s)),$$

where $H(\pi) = -\sum_a \pi(a|s) \log \pi(a|s)$ and $\beta > 0$ is the entropy coefficient. The well-known solution is the Boltzmann/softmax distribution:

$$\pi_\beta(a|s) = \frac{\exp(A(s, a)/\beta)}{\sum_{b \in \mathcal{A}} \exp(A(s, b)/\beta)}.$$

**Connection to $\epsilon$-greedy.**   Let $a^\star \in \arg \max_a A(s, a)$ and denote the advantage gaps $\Delta_a = A(s, a^\star) - A(s, a) \geq 0$. The probability of selecting the optimal action is

$$p_\star = \pi_\beta(a^\star|s) = \frac{1}{1 + \sum_{a \neq a^\star} \exp(-\Delta_a/\beta)}.$$

We can define a state- and value-dependent exploration probability $\epsilon_\beta(s) = 1 - p_\star$. This allows us to decompose the policy as:

$$\pi_\beta(\cdot|s) = (1 - \epsilon_\beta(s)) \, \delta_{a^\star} + \epsilon_\beta(s) \, q_\beta(\cdot|s),$$

where $q_\beta(a|s) \propto \exp(-\Delta_a/\beta)$ is a probability distribution over the set of suboptimal actions.

This formulation reveals that softmax exploration is a generalized form of $\epsilon$-greedy. However, unlike the standard $\epsilon$-greedy rule, its exploration is not uniform. The distribution $q_\beta$ intelligently assigns higher probability to suboptimal actions that are closer to optimal (i.e., having a smaller advantage gap $\Delta_a$). Only under the strong and often unrealistic condition that all suboptimal actions are equally bad ($\Delta_a \approx$ const. for $a \neq a^\star$) does $q_\beta$ approach a uniform distribution, making the strategy resemble standard $\epsilon$-greedy. Thus, entropy regularization typically leads to a more efficient exploration strategy than its uniform counterpart.

## C   ADDITION DESCRIPTIONS FOR OUR METHOD

Our algorithmic description for MERCI is as follows:

In the pipeline of bonus filtering, the step 3, i.e., *Noise-suppression Filtering*, can vary across tasks and can be optionally applied or configured depending on specific task requirements.

---

**Algorithm 1** Motivating Exploration in LLM Reasoning with Count-based Intrinsic Rewards

---

**Input:** policy model $\pi_\theta$, coin flipping network $f_\phi$, dataset $\mathcal{D}$, iterations $N$, outcome-based reward function $R$, exploration coefficient $\gamma$, clipping factor $\alpha$.

**for** $i = 1$ to $N$ **do**

    Generate $y \sim \pi_\theta(\cdot|x)$ for each prompt x in $\mathcal{D}_i$, and use $R$ to compute $\hat{A}_{\text{old}}$ via Equation 3.

    Extract hidden expression $s_{hidden}$ of each token in $y$ as described in Section 3.2.

    Compute $y$'s bonus with the process introduced in Section 3.3 and Figure 2, then incorporate it into the original advantage by applying $\gamma$ and $\alpha$ via Equation 8.

    Generate random vectors $c$ and update the parameter $\phi$ via Equation 1.

    Update the LLM policy $\pi_\theta$ using $\hat{A}_{\text{new}}$ in Equation 8.

**end for**

**Output:** Fine-tuned $\pi_\theta$ and $f_\phi$.

---

# D  PROOF OF PROPOSITION 1

The proof follows the methodology presented in (O'Donoghue et al., 2017). According to the definition of the conditional variance, we have:

$$
\mathbb{V}_t[\hat{Q}^h(s,a)] = \mathbf{E}_t\left[(\hat{Q}^h(s,a) - \mathbf{E}_t[\hat{Q}^h(s,a)])^2\right]
$$

$$
= \mathbf{E}_t\left[\left(\hat{r}^h(s) - \mathbf{E}_t[\hat{r}^h(s)] + \sum_{s',a'} \pi^h_{s'a'} P^h_{s'sa'}(\hat{Q}^{h+1}(s',a') - \mathbf{E}_t[\hat{Q}^{h+1}(s',a')])\right)^2\right]
$$

$$
= \mathbf{E}_t\left[(\hat{r}^h(s) - \mathbf{E}_t[\hat{r}^h(s)])^2\right]
$$

$$
+ \mathbf{E}_t\left[\left(\sum_{s',a'} \pi^h_{s'a'} P^h_{s'sa'}(\hat{Q}^{h+1}(s',a') - \mathbf{E}_t[\hat{Q}^{h+1}(s',a')])\right)^2\right]
$$

The second equality holds by expanding the square and assuming that the reward estimate $\hat{r}^h(s)$ and the next-step Q-value estimate $\hat{Q}^{h+1}(s',a')$ are conditionally independent, which makes their cross-product term zero.

Now, we focus on the second term. Noting that $\sum_{s',a'} \pi^h_{s'a'} P^h_{s'sa'} = 1$, this term represents a weighted sum. Since the function $f(x) = x^2$ is convex, we can apply **Jensen's inequality**.

For a convex function $f$, weights $w_i$ that sum to 1, and random variables $Z_i$, Jensen's inequality states:

$$
\mathbf{E}\left[f\left(\sum_i w_i Z_i\right)\right] \leq \sum_i w_i \mathbf{E}\left[f(Z_i)\right]
$$

Applying this to our expression gives:

$$
\mathbf{E}_t\left[\left(\sum_{s',a'} \pi^h_{s'a'} P^h_{s'sa'}(\hat{Q}^{h+1}(s',a') - \mathbf{E}_t[\hat{Q}^{h+1}(s',a')])\right)^2\right]
$$

$$
\leq \sum_{s',a'} \pi^h_{s'a'} P^h_{s'sa'}\, \mathbf{E}_t\left[(\hat{Q}^{h+1}(s',a') - \mathbf{E}_t[\hat{Q}^{h+1}(s',a')])^2\right]
$$

$$
= \sum_{s',a'} \pi^h_{s'a'} P^h_{s'sa'}\, \mathbb{V}_t[\hat{Q}^{h+1}(s',a')]
$$

Combining the results, we arrive at the final inequality:

$$
\mathbb{V}_t[\hat{Q}^h(s,a)] \leq \mathbb{V}_t[\hat{r}^h(s)] + \sum_{s',a'} \pi^h_{s'a'} P^h_{s'sa'}\, \mathbb{V}_t[\hat{Q}^{h+1}(s',a')]
$$

This shows that the variance of the Q-value at step $h$ is bounded by the variance of the immediate reward plus the expected variance of the Q-value at the next step, $h + 1$.

Table 3: Our CFN training configurations on mathematical reasoning tasks.

| Hyperparameter | Value |
|---|---|
| Optimizer | AdamW |
| Learning rate in the pretraining process | 1e-3 |
| Learning rate in the RL training process | 1e-4 |
| Training batch size | $512 \times 8$ |
| Mini-batch size | 8 |

## E   DETAILED TRAINING CONFIGURATIONS

### E.1   TRAINING DATA AND REWARD FUNCTION

**Mathematical Reasoning**    For both our train dataset and test dataset, we use the following system prompt:

> **System Prompt**
>
> Please reason step by step, and put your final answer within \boxed{}.

We use an outcome-based reward function that assigns +1 for correct final answers and -1 otherwise.

**SQL Generation**    For both our train dataset and test dataset, we do not explicitly use any system prompt. We add the following contents at the beginning of the user prompt:

> **Prompt**
>
> Task Overview:
> You are a data science expert. Below, you are provided with a database schema and a natural language question. Your task is to understand the schema and generate a valid SQL query to answer the question.

The outcome-based reward function is dense: final_score = answer_score + format_score, where:

$$
\text{answer\_score} = \begin{cases} 1.0, & \text{if } \text{Result}(S) = \text{Result}(G) \\ \min\left(\frac{\text{count}^2}{|\text{gold\_dict}| \times |\text{result\_dict}|}, 1.0\right) \times 0.8 & \text{if } \text{Result}(S) \neq \text{Result}(G) \end{cases} \tag{9}
$$

Above, $S$ is the generated solution string (predicted SQL query), and $G$ is the ground truth query. $\text{Result}(Q)$ is the set of execution results returned by the database when executing the SQL query $Q$.

### E.2   CFN TRAINING CONFIGURATION

For CFN training, we first generate rollouts from the backbone model on the training dataset and use these responses to perform a preliminary training of the CFN. This process enables it to develop a basic understanding of which states are likely to occur more rarely. During the RL training phase, we initialize the exploration model with the parameters of the pretrained CFN to prevent the information it provides at the outset from misleading the policy model.

**Mathematical Reasoning**    We use the hyperparameters in Table 3 for CFN training on mathematical reasoning tasks.

**SQL Generation**    We use the hyperparameters in Table 4 for CFN training on SQL generation tasks.

Table 4: Our CFN training configurations on SQL generation tasks.

| Hyperparameter | Value |
|---|---|
| Optimizer | AdamW |
| Learning rate in the pretraining process | 3e-4 |
| Learning rate in the RL training process | 1e-4 |
| Training batch size | $128 \times 8$ |
| Mini-batch size | 8 |

Table 5: Our RL training configurations on mathematical reasoning tasks.

| Hyperparameter | Value |
|---|---|
| *General training hyperparameters* | |
| Optimizer | AdamW |
| Policy learning rate | 1e-6 |
| Training batch size | 512 |
| Samples per prompt | 8 |
| Mini-batch size | 32 |
| Max prompt length | 1024 |
| Max response length | 3072 |
| Rollout temperature | 1.0 |
| *Method-specific hyperparameters* | |
| Top $p\%$ in step 1 of bonus filtering | 30% |
| Initial $\gamma$ in Equation 8 | 0.4 |
| $\alpha$ in Equation 8 | 0.5 |

### E.3 RL TRAINING CONFIGURATION

Our experiments were conducted on 32 NVIDIA H20-96GB GPUs. For the reproduced Entropy Adv. and iMentor methods, we adopt the same general training hyperparameters as listed in Table 5 and Table 6, while their method-specific hyperparameters follow the configurations reported in the original papers.

**Mathematical Reasoning** We use the hyperparameters in Table 5 for RL training on mathematical reasoning tasks. Notably, during RL training, we applied a cosine decay schedule to the discount factor $\gamma$, configured so that by step 200 it reached $10\%$ of its initial value. The same applies in the SQL generation task.

In addition to focusing on mean@$k$, we also place considerable emphasis on pass@$k$. However, we observe that as vanilla GRPO training progresses, increases in mean@$k$ are generally accompanied by sharp decreases in pass@$k$, which is also presented in Appendix G.2.2. Therefore, to ensure comparability across both types of metrics, we train each experiment for 120 steps on vanilla GRPO. For DAPO, we train each experiment for 160 steps (including data sampling and filtering).

**SQL Generation** We use the hyperparameters in Table 6 for RL training on SQL generation tasks.

We train each experiment for 160 steps on vanilla GRPO, and 240 steps on DAPO (including data sampling and filtering).

## F INFERENCE CONFIGURATIONS

**Mathematical Reasoning** We use a rollout temperature of $0.6$, top-$p$ sampling with $p = 0.95$, and a maximum response length of 4096 tokens. We adopt $k = 256$ for the small but challenging AIME2024/2025 datasets (30 problems each), $k = 16$ for Minerva, MATH500, and Olympiad-Bench, and $k = 8$ for College Math, balancing computational cost and difficulty.

Table 6: Our RL training configurations on SQL generation tasks.

| Hyperparameter | Value |
|---|---|
| *General training hyperparameters* | |
| Optimizer | AdamW |
| Policy learning rate | 1e-6 |
| Training batch size | 128 |
| Samples per prompt | 8 |
| Mini-batch size | 64 |
| Max prompt length | 8192 |
| Max response length | 4096 |
| Rollout temperature | 1.0 |
| *Method-specific hyperparameters* | |
| Top $p\%$ in step 1 of bonus filtering | 20% |
| Initial $\gamma$ in Equation 8 | 0.1 |
| $\alpha$ in Equation 8 | 0.5 |

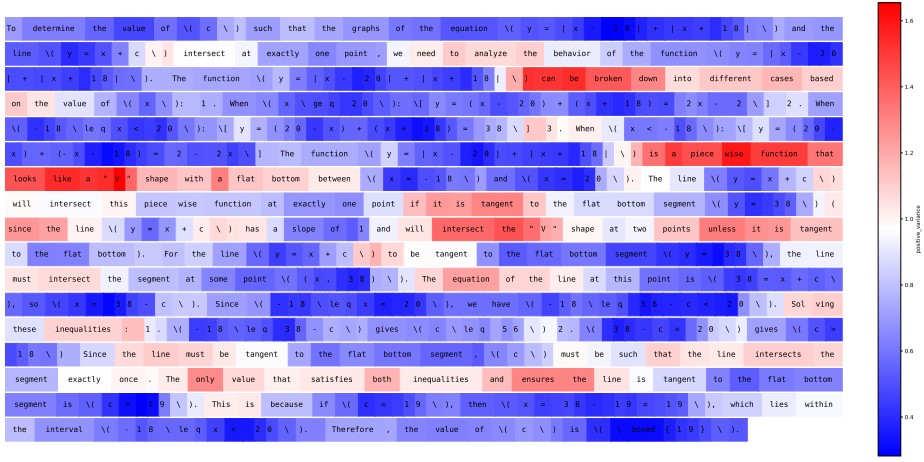

Figure 3: An example of token-level estimated epistemic uncertainty within a response. Red regions indicate relatively higher uncertainty estimates assigned by the CFN to the corresponding token positions, while blue regions indicate relatively lower estimates. The same applies hereafter.

**SQL Generation** We use a top-$p$ sampling with $p = 0.95$, and a maximum response length of 4096 tokens. We use a rollout temperature of $0.0$ for greedy sampling, and a rollout temperature of $1.0$ to evaluate pass@$k$.

# G ADDITIONAL EXPERIMENTAL RESULTS

## G.1 COIN FLIPPING NETWORK

**Uncertainty Estimation** We conduct two experiments to evaluate CFN as described in Section 5.2.1, and the detailed results are presented as follows in Figure 3, 4, 5, 6 and Figure 7.

**Generalization Ability of CFN** We directly apply the CFN trained on mathematical reasoning tasks to estimate the uncertainty of responses in the SQL Generation task, and some examples are shown in Figure 8, Figure 9 and Figure 10. Since SQL code is indeed infrequently encountered, it exhibits higher uncertainty, and the CFN correspondingly produces noticeably elevated estimates. In contrast, the uncertainty values assigned to other natural language reasoning sequences are largely consistent with our intuition and expectations. This directly demonstrates that the CFN is capable of

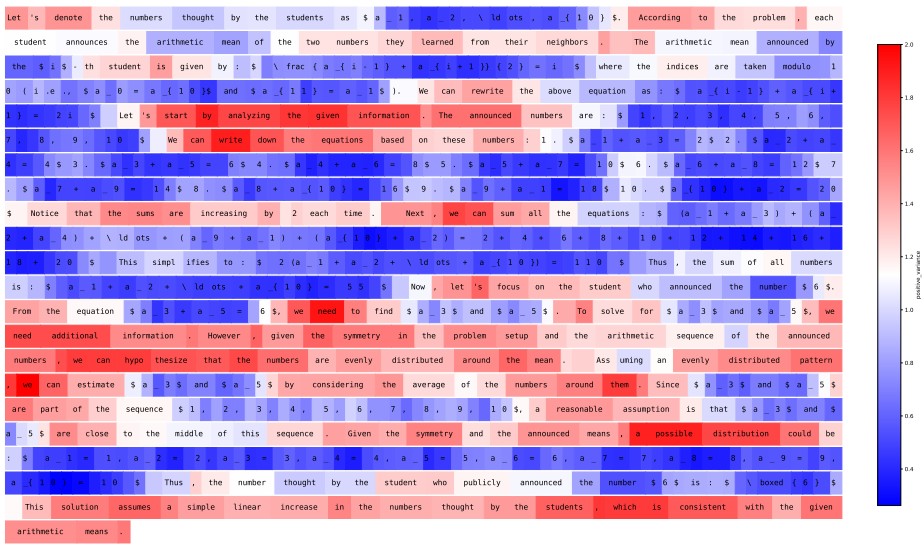

Figure 4: An example of token-level estimated epistemic uncertainty within a response.

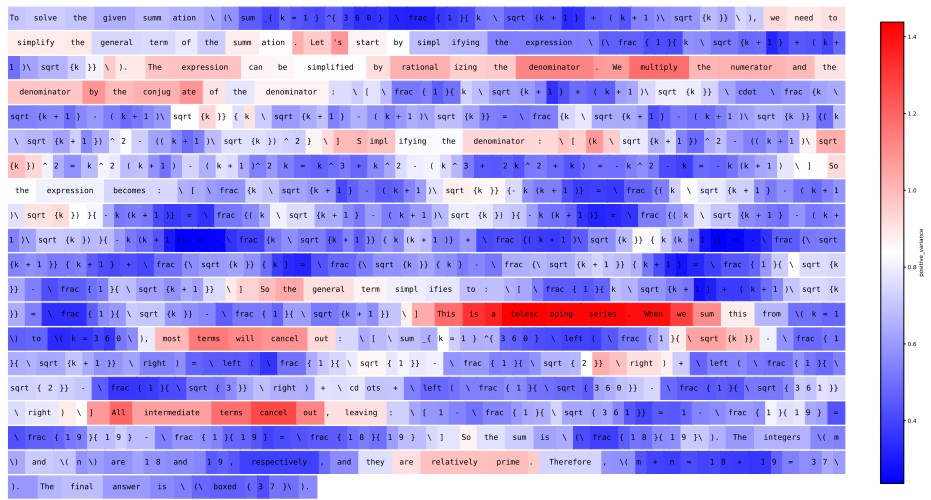

Figure 5: An example of token-level estimated epistemic uncertainty within a response.

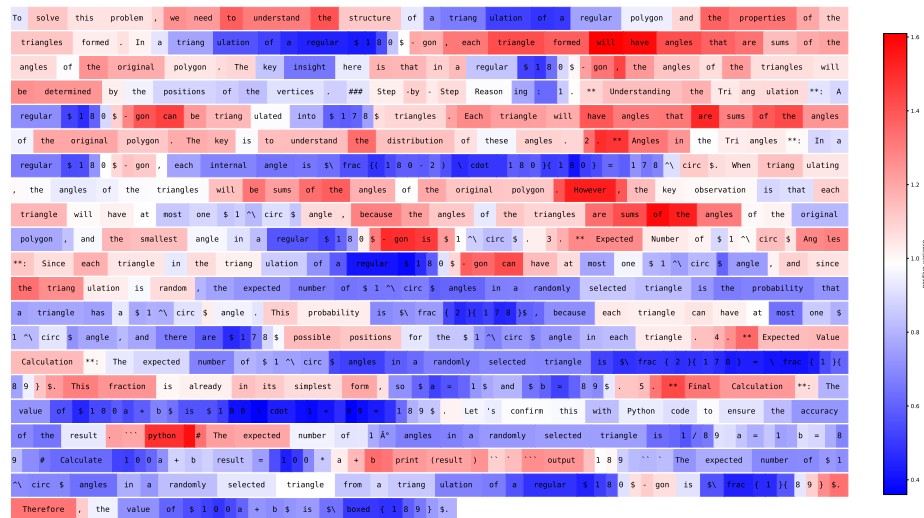

Figure 6: An example of token-level estimated epistemic uncertainty within a response.

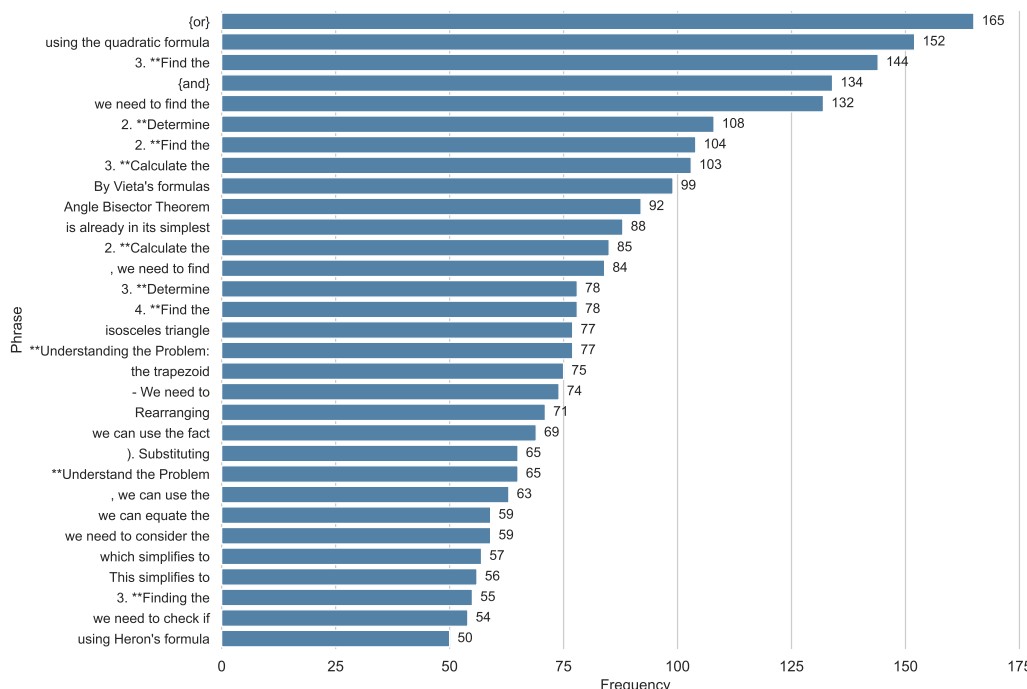

Figure 7: A statistical analysis of the occurrence frequency of contiguous token segments within each response that fall within the top 30% of bonus values (after filtering out code-related segments).

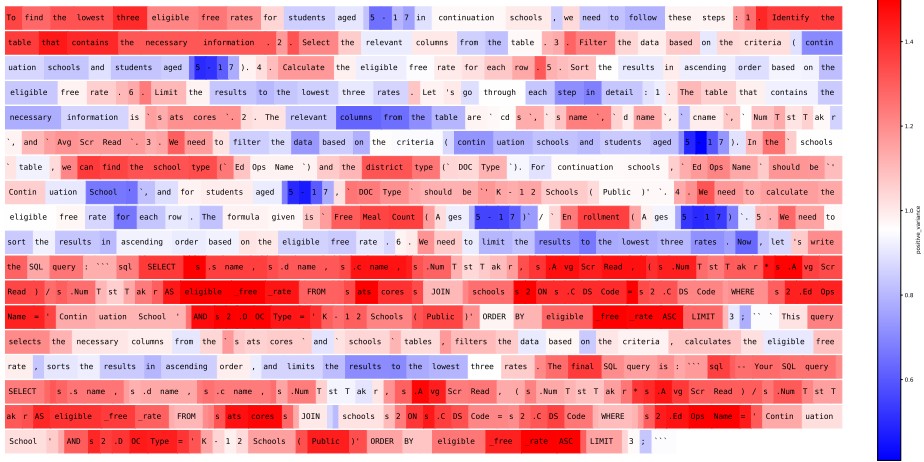

Figure 8: An example of token-level estimated epistemic uncertainty within a response for the SQL generation task. The CFN is trained on mathematical reasoning tasks.

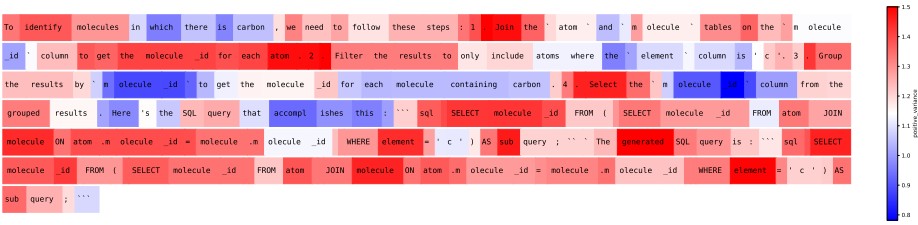

Figure 9: An example of token-level estimated epistemic uncertainty within a response for the SQL generation task. The CFN is trained on mathematical reasoning tasks.

leveraging the LLM's general features and translating them into an effective novelty estimate across domains.

**Semantic Capturing**  To assess the CFN's ability to capture semantic similarity and to determine whether it can provide reasonable uncertainty estimates for sequences that are linguistically close but not identical, we present in Figure 11 several example statements along with their corresponding aggregated trajectory uncertainties. The results are obtained from the CFN at training step 80.

The underlying rationale and semantics of (1) and (2) are linguistically close but not identical, and the resulting overall uncertainty estimates for the trajectories are also close. In contrast, (1) and

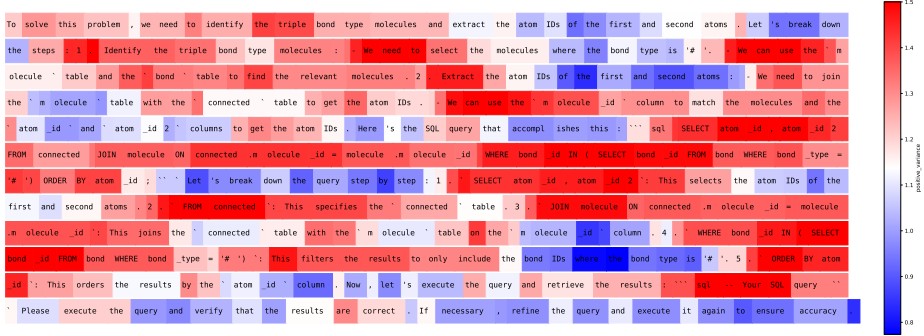

Figure 10: An example of token-level estimated epistemic uncertainty within a response for the SQL generation task. The CFN is trained on mathematical reasoning tasks.

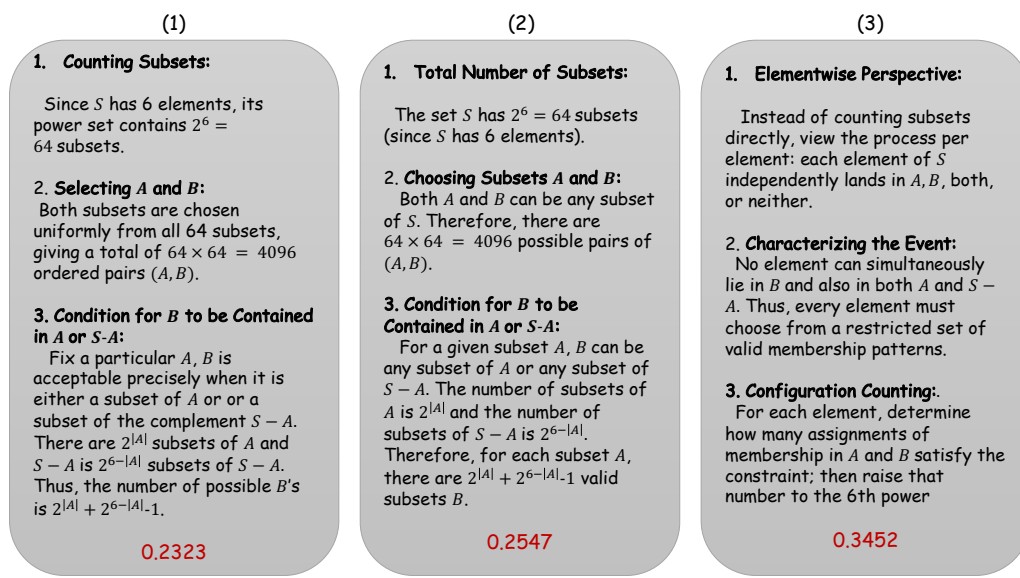

Figure 11: An example of reasoning trajectories and corresponding aggregated uncertainties.

(3) differ more substantially, leading to larger discrepancies in the trajectory-aggregated uncertainty estimates.

**Relation with Entropy, Score and Rollouts Variance** We record the following information during the process of policy model rollout: (a) the CFN uncertainty estimates and policy entropy at each token position; (b) the average CFN uncertainty estimate for each trajectory and the corresponding within-group rollout variance; and (c) the average CFN uncertainty estimate for each trajectory and the final score (i.e., empirical success rate). The scatter of these records are shown in Figure 12.

These results indicate that: (a) Positions at which CFN predicts higher uncertainty do not necessarily correspond to higher entropy. Policy entropy measures the policy's action randomness, whereas the CFN is designed to measure the model's epistemic uncertainty regarding its own state knowledge. Thus, our results confirm that the CFN provides a unique, non-redundant signal that cannot be simply replaced by the policy's action probabilities. (b) For trajectories that ultimately fail, the average uncertainty estimated by the CFN tends to be higher. This aligns with our understanding, since highly novel states typically correspond to regions that the model has insufficiently explored. In such under-explored regions, the probability of ultimately solving the problem should naturally be lower. (c) In terms of trajectory uncertainty and the variance of the corresponding within-group rollout results, no particularly pronounced correlation is observed. Some trajectories with zero rollout variance exhibit relatively high uncertainty, which may be due to the fact that we generate rollouts using the base model; in datasets of relatively higher difficulty, this can lead to a larger number of entirely incorrect trajectories.

## G.2 RL Training

### G.2.1 Cross-domain Experiments

To further evaluate the cross-domain effectiveness of our approach, we transfer the RL models trained on the Mathematical Reasoning dataset to downstream tasks such as MMLU-Pro and GPQA for testing. We convert each problem into a multiple-choice question (MCQ) format, and the system prompt is as follows. For GPQA, we sample up to 16 times, whereas for MMLU-Pro, we sample only once due to its large scale. The results are shown in Table 7.

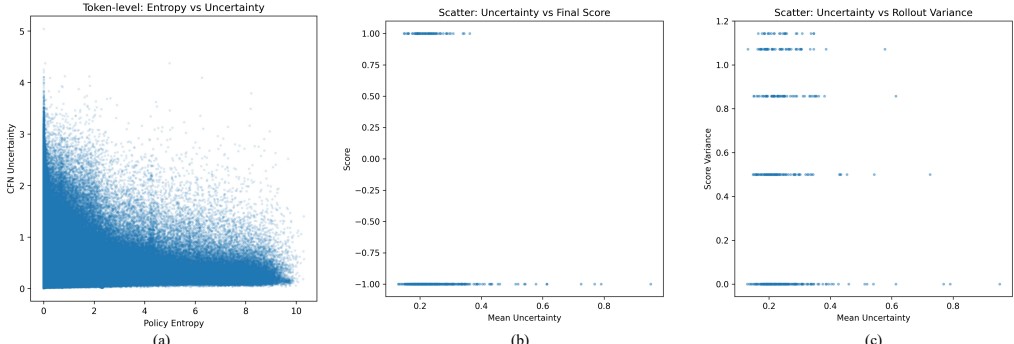

Figure 12: Relations between the CFN Uncertainty Esitimation and Entropy, Score and Rollouts Variance.

Table 7: Results of cross-domain experiments on MMLU-Pro and GPQA.

| | GPQA | | | MMLU-Pro |
| | mean@16 | pass@8 | pass@16 | pass@1 |
|---|---|---|---|---|
| *Qwen2.5-Math* | 8.98 | 47.5 | 53.6 | 5.8 |
| + GRPO | 24.3 | 59.8 | 61.6 | 28.3 |
| + GRPO w / MERCI | 26.4 | 65.0 | 69.6 | 29.1 |
| + DAPO | 26.2 | 61.2 | 70.5 | 37.4 |
| + DAPO w / MERCI | 27.4 | 64.3 | 73.7 | 39.5 |

---

**System Prompt**

What of the following is the right choice? Please reason step by step, and put your final answer within \boxed{}. The final answer must be a capital letter like A, B, C, or D.

---

The results show that incorporating MERCI consistently provides gains on top of both GRPO and DAPO, particularly on GPQA pass@8 and pass@16, as well as MMLU-Pro pass@1. These findings suggest that our MERCI play an important role in improving out-of-domain robustness, even when the underlying training data is highly domain-specialized.

### G.2.2    SCALING EXPERIMENTS

As a comparison, we scaled the vanilla GRPO baseline experiment, continuing to train the GRPO baseline to the $260th$ training step. We observe that, although extended training increases mean@$k$, it substantially degrades pass@$k$, consistent with the limitations of GRPO discussed in Appendix E.3. By contrast, our MERCI algorithm realizes its exploratory potential earlier, rapidly identifying good solutions and exhibiting improvements in pass@$k$ as well.

### G.2.3    TRAINING DYNAMICS

We report the training dynamics of both validation accuracy and response length on mathematical reasoning tasks in Figure 13.

### G.2.4    ABLATION STUDIES

To verify the effectiveness of the modules in our method, we conducted ablation studies on the mathematical reasoning task and vanilla GRPO, and present the results in Table 9.

As evidenced by the preceding experimental results, both noise filtering and normalized trajectory-aggregated uncertainty estimation are critical to our method; without them, training can become

Table 8: Results of scaling experiments on vanilla GRPO and mathematical reasoning benchmarks.

(a) pass@k results

|  | AIME25 pass@256 | AIME24 pass@256 | Minerva pass@16 | MATH500 pass@16 | OlympiadBench pass@16 | College pass@8 | Avg. |
|---|---|---|---|---|---|---|---|
| GRPO | 50.0 | 76.7 | 64.0 | 91.8 | 59.7 | 49.2 | 65.8 |
| GRPO-scaling180 | 50.0 | 66.7 | 61.4 | 90.2 | 57.3 | 48.2 | 62.3 |
| GRPO-scaling260 | 46.7 | 70.0 | 59.6 | 88.8 | 56.4 | 48.1 | 61.6 |
| GRPO + MERCI | 60.0 | 80.0 | 63.2 | 91.4 | 60.9 | 48.9 | 67.4 |
| GRPO + MERCI-scaling180 | 56.7 | 73.3 | 61.4 | 90.4 | 58.7 | 48.5 | 64.8 |
| GRPO + MERCI-scaling260 | 50.0 | 70.0 | 61.2 | 89.4 | 58.5 | 47.5 | 62.8 |

(b) mean@k results

|  | AIME25 mean@256 | AIME24 mean@256 | Minerva mean@16 | MATH500 mean@16 | OlympiadBench mean@16 | College mean@8 | Avg. |
|---|---|---|---|---|---|---|---|
| GRPO | 11.2 | 28.7 | 41.8 | 79.0 | 40.3 | 42.0 | 40.5 |
| GRPO-scaling180 | 13.1 | 27.1 | 42.0 | 78.9 | 41.2 | 42.9 | 40.9 |
| GRPO-scaling260 | 12.7 | 28.3 | 42.8 | 78.7 | 40.8 | 42.7 | 41.0 |
| GRPO + MERCI | 13.4 | 29.6 | 44.1 | 80.7 | 42.6 | 42.9 | 42.2 |
| GRPO + MERCI-scaling180 | 14.1 | 31.7 | 43.0 | 80.7 | 41.9 | 42.5 | 42.3 |
| GRPO + MERCI-scaling260 | 12.9 | 30.9 | 43.6 | 80.6 | 42.5 | 43.0 | 42.3 |

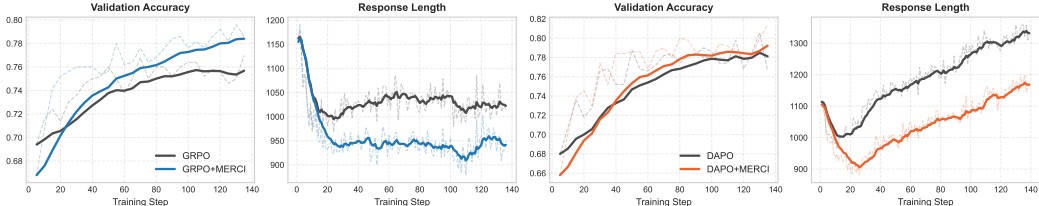

Figure 13: Validation (i.e., MATH500) accuracy and response length during training.

unstable and may even underperform the baseline algorithm. Furthermore, percentile and spatial coherence filtering direct attention to uncertainty at key positions, while the variance-accumulation method yields more accurate estimates, thereby further improving our algorithm's performance.

### G.2.5 HYPERPARAMETER CHOICES

**Dimension of CFN**  As a comparison, we set the dimension $d$ of CFN to 40, and the experimental results are shown in Figure 14. We posit that if the CFN dimension were significantly larger, the high-dimensional LLM hidden states, when projected through the network head, might experience reduced distinguishability. A large dimension could potentially oversmooth the feature space, causing subtle but important differences between novel and seen states to become less pronounced.

**Decay of the exploration coefficient**  To discuss how the slower/faster cosine schedule decay impacts the final performance, we conducted experiments with the decay steps set to 100, 200, and 300, respectively, and the results are presented in Table 10.

The experimental results demonstrate that the cosine decay of $\gamma$ to 10% by step 200 provides better overall performance. Schedules with a faster decay rate (e.g., decaying to 10% by step 100) led to insufficient exploration time. The policy quickly stabilized into suboptimal reasoning paths, resulting in a lower ceiling for the final performance. Conversely, schedules with a slower decay (e.g., decaying by step 300) hindered convergence late in training. The persistent, strong intrinsic reward introduced excessive noise or bias, preventing the policy from focusing on maximizing the external task reward, thus degrading the final performance and stability.

Table 9: Ablation studies on vanilla GRPO and mathematical reasoning benchmarks. *p & s filtering* is a reduced-form representation of percentile and spatial coherence filtering. **The difference between *cumulative std* and *cumulative variance* (i.e, our main results) has been introduced in Paragraph 1 of Section 3.3. † The setting of token integration expresses that, rather than computing uncertainty at the trajectory level for entire trajectories, we directly add an uncertainty estimate to each token-level advantage. ‡ MERCI w/o normalization is the results of strictly following the sum-then-sqrt computation and divide the result by a constant factor (set to 1000), without applying any additional normalization.

(a) pass@$k$ results

| | AIME25 pass@256 | AIME24 pass@256 | Minerva pass@16 | MATH500 pass@16 | OlympiadBench pass@16 | College pass@8 | Avg. |
|---|---|---|---|---|---|---|---|
| GRPO | 53.3 | 76.7 | 64.0 | 91.8 | 59.7 | 49.2 | 65.8 |
| GRPO + MERCI w/o *p & s filtering** | 56.7 | 73.3 | 65.8 | 90.0 | 60.0 | 47.9 | 65.6 |
| GRPO + MERCI w/o noise filtering | 50.0 | 73.3 | 61.8 | 89.2 | 59.7 | 48.9 | 63.8 |
| GRPO + MERCI w/o normalization ‡ | 53.3 | 76.7 | 63.6 | 89.8 | 58.7 | 48.7 | 65.1 |
| GRPO + MERCI w/ *cumulative std** | 56.7 | 76.7 | 65.1 | 90.6 | 60.2 | 48.1 | 66.2 |
| GRPO + MERCI w/ token integration† | 46.7 | 70.0 | 62.5 | 91.6 | 59.3 | 49.1 | 63.2 |
| GRPO + MERCI | 60.0 | 80.0 | 63.2 | 91.4 | 60.9 | 48.9 | 67.4 |

(b) mean@$k$ results

| | AIME25 mean@256 | AIME24 mean@256 | Minerva mean@16 | MATH500 mean@16 | OlympiadBench mean@16 | College mean@8 | Avg. |
|---|---|---|---|---|---|---|---|
| GRPO | 11.2 | 28.7 | 41.8 | 79.0 | 40.3 | 42.0 | 40.5 |
| GRPO + MERCI w/o *p & s filtering* | 11.7 | 28.1 | 44.9 | 79.9 | 39.9 | 42.6 | 41.2 |
| GRPO + MERCI w/o noise filtering | 9.8 | 25.8 | 40.6 | 77.3 | 37.9 | 41.9 | 38.9 |
| GRPO + MERCI w/o normalization | 12.4 | 29.8 | 44.0 | 80.0 | 40.4 | 42.6 | 41.5 |
| GRPO + MERCI w/ *cumulative std* | 14.2 | 29.1 | 43.8 | 79.8 | 41.2 | 43.0 | 41.9 |
| GRPO + MERCI w/ token integration | 12.0 | 23.7 | 40.2 | 77.5 | 39.9 | 42.4 | 39.3 |
| GRPO + MERCI | 13.4 | 29.6 | 44.1 | 80.7 | 42.6 | 42.9 | 42.2 |

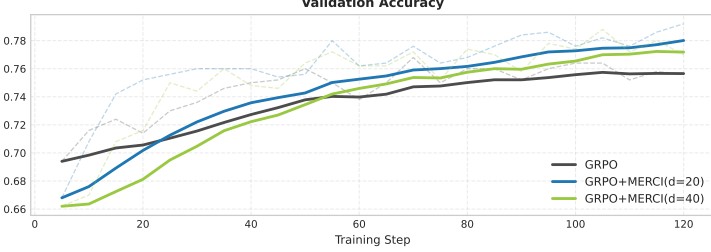

Figure 14: Validation (i.e., MATH500) accuracy during training across different choices on the dimension of CFN.

**Top-p% Used in Filtering**   For the percentile values used in the filtering step (i.e., top $p\%$), we likewise conducted experiments with settings of 20%, 30%, and 50%, respectively, and the results are shown in Figure 15.

### G.2.6   CASE STUDY

We analyzed two cases on AIME2024/AIME2025 to examine the effect of incorporating our method, and the results are as follows.

In Case Study 1 on AIME2024, compared with the DAPO solution, our DAPO+MERCI method provides a significantly clearer and more rigorous derivation. By organizing the substitutions into a structured sequence and isolating each variable through simple exponential equations, it avoids

Table 10: Results of the speed of cosine schedule decay.

(a) pass@$k$ results

| | AIME25 pass@256 | AIME24 pass@256 | Minerva pass@16 | MATH500 pass@16 | OlympiadBench pass@16 | College pass@8 | Avg. |
|---|---|---|---|---|---|---|---|
| GRPO + MERCI-decaystep200 | 60.0 | 80.0 | 63.2 | 91.4 | 60.9 | 48.9 | 67.4 |
| GRPO + MERCI-decaystep100 | 56.7 | 80.0 | 62.1 | 90.6 | 61.3 | 48.6 | 66.6 |
| GRPO + MERCI-decaystep300 | 56.7 | 76.7 | 62.9 | 90.8 | 60.3 | 48.7 | 66.0 |

(b) mean@$k$ results

| | AIME25 mean@256 | AIME24 mean@256 | Minerva mean@16 | MATH500 mean@16 | OlympiadBench mean@16 | College mean@8 | Avg. |
|---|---|---|---|---|---|---|---|
| GRPO + MERCI-decaystep200 | 13.4 | 29.6 | 44.1 | 80.7 | 42.6 | 42.9 | 42.2 |
| GRPO + MERCI-decaystep100 | 12.9 | 28.9 | 43.8 | 80.0 | 43.1 | 43.4 | 42.0 |
| GRPO + MERCI-decaystep300 | 14.4 | 27.0 | 41.3 | 80.6 | 41.5 | 42.4 | 41.2 |

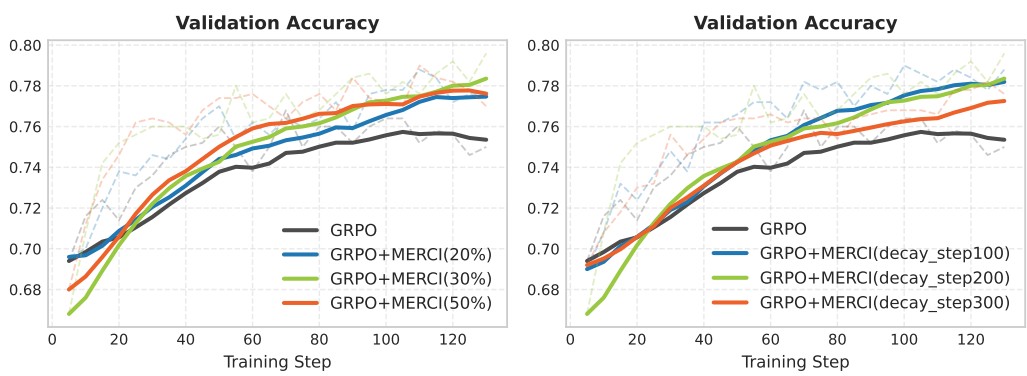

Figure 15: Validation (i.e., MATH500) accuracy during training across different choices on top $p\%$ and cosine decay step.

the excessive algebraic accumulation seen in the DAPO approach. The resulting argument is more transparent, mathematically systematic, and pedagogically effective.

In Case Study 2 on AIME2025, the DAPO baseline's solution relies on extensive algebraic expansion followed by heuristic symmetry arguments that are not rigorously justified, causing a disconnect between computation and conclusion. In contrast, our DAPO+MERCI method's solution adopts a structurally motivated approach based on logarithmic differentiation, which directly relates the locations of the roots to the behavior of the critical points. Although more concise, it provides a coherent analytic framework and a logically valid path to the correct values.

In summary, these studies consistently validate the observation that the incorporation of the uncertainty-driven intrinsic reward fundamentally alters the policy model's approach to multi-step reasoning: **1. Enhanced Strategy Coherence and Conciseness**: CFN's novelty mechanism cuts out "cognitive noise" and long, self-correcting detours, leading to significantly improved strategy coherence and sample efficiency; **2. Cognitive Migration to Higher-Level Reasoning**: The increased proportion of steps devoted to higher-level reasoning is the most significant indicator of improved cognitive efficiency; **3. Discovery of Novel and Elegant Solutions**: The provided case study example also demonstrates that the MERCI-guided policy finds more elegant, efficient, or fundamentally different mathematical/structural approaches to solving problems.

**Case Study 1 on AIME2024**

**Question:** Let $x, y$ and $z$ be positive real numbers that satisfy the following system of equations:

$log_2\left(\frac{x}{yz}\right) = \frac{1}{2}$   $log_2\left(\frac{y}{xz}\right) = \frac{1}{3}$   $log_2\left(\frac{z}{xy}\right) = \frac{1}{4}$

Then the value of $\left|log_2(x^4y^3z^2)\right|$ is $\frac{m}{n}$ where $m$ and $n$ are relatively prime positive integers. Find $m+n$.

**Response (DAPO):** Let's solve the given system of equations step by step. The system of equations provided is:

$$\log_2\left(\frac{x}{yz}\right) = \frac{1}{2}$$

$$\log_2\left(\frac{y}{xz}\right) = \frac{1}{3}$$

$$\log_2\left(\frac{z}{xy}\right) = \frac{1}{4}$$

First, we can rewrite each logarithmic equation in exponential form:1. $\log_2\left(\frac{x}{yz}\right) = \frac{1}{2}$ implies $\frac{x}{yz} = 2^{\frac{1}{2}} = \sqrt{2}$, so $x = \sqrt{2} \cdot yz$.2. $\log_2\left(\frac{y}{xz}\right) = \frac{1}{3}$ implies $\frac{y}{xz} = 2^{\frac{1}{3}}$, so $y = 2^{\frac{1}{3}} \cdot xz$.3. $\log_2\left(\frac{z}{xy}\right) = \frac{1}{4}$ implies $\frac{z}{xy} = 2^{\frac{1}{4}}$, so $z = 2^{\frac{1}{4}} \cdot xy$.

Now, we can substitute the expressions for $x$, $y$, and $z$ into each other to find the values of $x$, $y$, and $z$:

From the first equation: $x = \sqrt{2} \cdot yz$

From the second equation: $y = 2^{\frac{1}{3}} \cdot xz$

From the third equation: $z = 2^{\frac{1}{4}} \cdot xy$

By multiplying these three equations together:

$$x \cdot y \cdot z = (\sqrt{2} \cdot yz) \cdot (2^{\frac{1}{3}} \cdot xz) \cdot (2^{\frac{1}{4}} \cdot xy)$$

$$x \cdot y \cdot z = \sqrt{2} \cdot 2^{\frac{1}{3}} \cdot 2^{\frac{1}{4}} \cdot (yz \cdot xz \cdot xy)$$

$$x \cdot y \cdot z = \sqrt{2} \cdot 2^{\frac{1}{3}} \cdot 2^{\frac{1}{4}} \cdot (x^2y^2z^2)$$

$$x \cdot y \cdot z = 2^{\frac{1}{2}} \cdot 2^{\frac{1}{3}} \cdot 2^{\frac{1}{4}} \cdot (x^2y^2z^2)$$

$$x \cdot y \cdot z = 2^{\frac{1}{2}+\frac{1}{3}+\frac{1}{4}} \cdot (x^2y^2z^2)$$

$$x \cdot y \cdot z = 2^{\frac{6}{12}+\frac{4}{12}+\frac{3}{12}} \cdot (x^2y^2z^2)$$

$$x \cdot y \cdot z = 2^{\frac{13}{12}} \cdot (x^2y^2z^2)$$

$$x \cdot y \cdot z = 2^{\frac{13}{12}} \cdot (x^2y^2z^2)$$

$$1 = 2^{\frac{13}{12}} \cdot (xyz)$$

$$1 = 2^{\frac{13}{12}} \cdot (xyz)$$

$$xyz = 2^{-\frac{13}{12}}$$

Now we can use $x = \sqrt{2} \cdot yz$, $y = 2^{\frac{1}{3}} \cdot xz$, and $z = 2^{\frac{1}{4}} \cdot xy$ to find the individual values of $x$, $y$, and $z$:From $x = \sqrt{2} \cdot yz$:

$$x = \sqrt{2} \cdot yz$$

$$x = \sqrt{2} \cdot \left(2^{-\frac{13}{12} \cdot \frac{1}{x}}\right)$$

Let's find the exponents of $x$, $y$, and $z$ in the expression $x = \sqrt{2} \cdot yz$:

$$x = \sqrt{2} \cdot yz$$

$$x = 2^{\frac{1}{2}} \cdot yz$$

$$x^2 = 2 \cdot y^2z^2$$

We know $xyz = 2^{-\frac{13}{12}}$, so:

$$x = 2^{\frac{1}{2}} \cdot \frac{2^{-\frac{13}{12}}}{x}$$
$$x^2 = 2^{\frac{1}{2}} \cdot 2^{-\frac{13}{12}}$$
$$x^2 = 2^{\frac{1}{2} - \frac{13}{12}}$$
$$x^2 = 2^{\frac{6}{12} - \frac{13}{12}}$$
$$x^2 = 2^{-\frac{7}{12}}$$
$$x = 2^{-\frac{7}{24}}$$

Similarly:

$$y = 2^{\frac{1}{3}} \cdot xz$$
$$y = 2^{\frac{1}{3}} \cdot \frac{2^{-\frac{13}{12}}}{y}$$
$$y^2 = 2^{\frac{1}{3}} \cdot 2^{-\frac{13}{12}}$$
$$y^2 = 2^{\frac{1}{3} - \frac{13}{12}}$$
$$y^2 = 2^{\frac{4}{12} - \frac{13}{12}}$$
$$y^2 = 2^{-\frac{9}{12}}$$
$$y^2 = 2^{-\frac{3}{4}}$$
$$y = 2^{-\frac{3}{8}}$$

Similarly:

$$z = 2^{\frac{1}{4}} \cdot xy$$
$$z = 2^{\frac{1}{4}} \cdot 2^{-\frac{7}{24}} \cdot 2^{-\frac{3}{8}}$$
$$z = 2^{\frac{1}{4}} \cdot 2^{-\frac{7}{24} - \frac{3}{8}}$$
$$z = 2^{\frac{1}{4}} \cdot 2^{-\frac{7}{24} - \frac{9}{24}}$$
$$z = 2^{\frac{1}{4}} \cdot 2^{-\frac{16}{24}}$$
$$z = 2^{\frac{1}{4}} \cdot 2^{-\frac{2}{3}}$$
$$z = 2^{\frac{3}{12} - \frac{8}{12}}$$
$$z = 2^{-\frac{5}{12}}$$

Now we need to find $\left|\log_2(x^4 y^3 z^2)\right|$:

$$x^4 y^3 z^2 = \left(2^{-\frac{7}{24}}\right)^4 \cdot \left(2^{-\frac{3}{8}}\right)^3 \cdot \left(2^{-\frac{5}{12}}\right)^2$$

$$x^4 y^3 z^2 = 2^{-\frac{7}{6}} \cdot 2^{-\frac{9}{8}} \cdot 2^{-\frac{5}{6}}$$
$$x^4 y^3 z^2 = 2^{-\frac{7}{6} - \frac{9}{8} - \frac{5}{6}}$$
$$x^4 y^3 z^2 = 2^{-\frac{56}{48} - \frac{54}{48} - \frac{40}{48}}$$
$$x^4 y^3 z^2 = 2^{-\frac{150}{48}} = 2^{-\frac{25}{8}}$$

Now we find the logarithm:

$$\left|\log_2(x^4 y^3 z^2)\right| = \left|\log_2\left(2^{-\frac{25}{8}}\right)\right| = \left|-\frac{25}{8}\right| = \frac{25}{8}$$

Thus, $m = 25$ and $n = 8$, so $m + n = 33$.
Let's confirm with Python:

**Python Code**

```python
# Calculating the valueimport math

# Values found

x_val = 2**(-7/24)y_val = 2**(-3/8)

z_val = 2**(-5/12)

# Calculate the expression x^4 * y^3 * z^2

expression_value = x_val**4 * y_val**3 * z_val**2

# Calculate log2 of the expression value

log_value = math.log2(expression_value)

# Absolute value

abs_log_value = abs(log_value)

print(abs_log_value)
```

**Output**

3.125

The calculation confirms that $\left|\log_2(x^4 y^3 z^2)\right| = \frac{25}{8} = 3.125$, which matches our derived value. Thus, $m = 25$ and $n = 8$, so $m + n = 33$.
Therefore, the answer is:

$$\boxed{33}$$

**Response (DAPO+MERCI):** To solve the given system of equations, we start by rewriting the logarithmic equations in exponential form:

1. $\log_2\left(\frac{x}{yz}\right) = \frac{1}{2}$
2. $\log_2\left(\frac{y}{xz}\right) = \frac{1}{3}$
3. $\log_2\left(\frac{z}{xy}\right) = \frac{1}{4}$

Using the definition of logarithms, we can convert these into the following equations:

1. $\frac{x}{yz} = 2^{\frac{1}{2}} = \sqrt{2}$
2. $\frac{y}{xz} = 2^{\frac{1}{3}} = \sqrt[3]{2}$
3. $\frac{z}{xy} = 2^{\frac{1}{4}} = \sqrt[4]{2}$

From these equations, we get:

$$x = \sqrt{2} \cdot yz \tag{1}$$
$$y = \sqrt[3]{2} \cdot xz \tag{2}$$
$$z = \sqrt[4]{2} \cdot xy \tag{3}$$

Now we substitute equation (1) into equation (2):

$$y = \sqrt[3]{2} \cdot (\sqrt{2} \cdot yz)z = \sqrt[3]{2} \cdot \sqrt{2} \cdot yz^2 = 2^{\frac{1}{3}+\frac{1}{2}} \cdot yz^2 = 2^{\frac{5}{6}} \cdot yz^2$$

Dividing both sides by $y$, we get:

$$1 = 2^{\frac{5}{6}} \cdot z^2$$

So,

$$z^2 = 2^{-\frac{5}{6}}$$

$$z = 2^{-\frac{5}{12}} \tag{4}$$

Now we substitute equation (1) into equation (3):

$$z = \sqrt[4]{2} \cdot (\sqrt{2} \cdot yz)y = \sqrt[4]{2} \cdot \sqrt{2} \cdot y^2 z = 2^{\frac{1}{4}+\frac{1}{2}} \cdot y^2 z = 2^{\frac{3}{4}} \cdot y^2 z$$

Dividing both sides by $z$, we get:

$$1 = 2^{\frac{3}{4}} \cdot y^2$$

So,

$$y^2 = 2^{-\frac{3}{4}}$$

$$y = 2^{-\frac{3}{8}} \tag{5}$$

Now we substitute equation (2) into equation (3):

$$z = \sqrt[4]{2} \cdot (\sqrt[3]{2} \cdot xz)x = \sqrt[4]{2} \cdot \sqrt[3]{2} \cdot x^2 z = 2^{\frac{1}{4}+\frac{1}{3}} \cdot x^2 z = 2^{\frac{7}{12}} \cdot x^2 z$$

Dividing both sides by $z$, we get:

$$1 = 2^{\frac{7}{12}} \cdot x^2$$

So,

$$x^2 = 2^{-\frac{7}{12}}$$

$$x = 2^{-\frac{7}{24}} \tag{6}$$

Now that we have the values:

$$x = 2^{-\frac{7}{24}}, \quad y = 2^{-\frac{3}{8}}, \quad z = 2^{-\frac{5}{12}}$$

We need to find the value of $\left|\log_2(x^4 y^3 z^2)\right|$.

First, we calculate $x^4 y^3 z^2$:

$$x^4 = \left(2^{-\frac{7}{24}}\right)^4 = 2^{-\frac{7}{6}}$$

$$y^3 = \left(2^{-\frac{3}{8}}\right)^3 = 2^{-\frac{9}{8}}$$

$$z^2 = \left(2^{-\frac{5}{12}}\right)^2 = 2^{-\frac{5}{6}}$$

So,

$$x^4 y^3 z^2 = 2^{-\frac{7}{6}} \cdot 2^{-\frac{9}{8}} \cdot 2^{-\frac{5}{6}} = 2^{-\frac{7}{6}-\frac{9}{8}-\frac{5}{6}} = 2^{-\left(\frac{7}{6}+\frac{9}{8}+\frac{5}{6}\right)}$$

To add the exponents, we need a common denominator:

$$\frac{7}{6} + \frac{9}{8} + \frac{5}{6} = \frac{56}{48} + \frac{54}{48} + \frac{40}{48} = \frac{56+54+40}{48} = \frac{150}{48} = \frac{25}{8}$$

So,

$$x^4 y^3 z^2 = 2^{-\frac{25}{8}}$$

Then,

$$\log_2(x^4 y^3 z^2) = \log_2\left(2^{-\frac{25}{8}}\right) = -\frac{25}{8}$$

Thus,

$$\left|\log_2(x^4 y^3 z^2)\right| = \left|-\frac{25}{8}\right| = \frac{25}{8}$$

The fraction $\frac{25}{8}$ is already in its simplest form, so $m = 25$ and $n = 8$. Therefore, $m + n = 25 + 8 = 33$.

The final answer is:

$$\boxed{33}$$

---

**Case Study 2 on AIME 2025**

**Question:** There are exactly three positive real numbers $k$ such that the function

$$f(x) = \frac{(x-18)(x-72)(x-98)(x-k)}{x}$$

defined over the positive real numbers achieves its minimum value at exactly two positive real numbers $x$. Find the sum of these three values of $k$.

- - - - - - - - - - - - - - - - - - - - - - - - - - - - - - - - - - - - - - - - - -

**Response (DAPO):** To solve this problem, we need to analyze the function $f(x) = \frac{(x-18)(x-72)(x-98)(x-k)}{x}$ and find the values of $k$ such that the function achieves its minimum value at exactly two positive real numbers $x$.First, let's rewrite $f(x)$ in a more manageable form:

$$f(x) = \frac{(x-18)(x-72)(x-98)(x-k)}{x}.$$

To find the critical points of $f(x)$, we need to find the derivative $f'(x)$ and set it to zero. Let's define:

$$g(x) = (x-18)(x-72)(x-98)(x-k).$$

Then,

$$f(x) = \frac{g(x)}{x}.$$

Using the quotient rule to differentiate $f(x)$:

$$f'(x) = \frac{g'(x) \cdot x - g(x) \cdot 1}{x^2} = \frac{xg'(x) - g(x)}{x^2}.$$

We need to find $g(x)$ and its derivative $g'(x)$:

$$g(x) = (x-18)(x-72)(x-98)(x-k).$$

To find $g'(x)$, we can use the product rule:

$$g'(x) = (x-72)(x-98)(x-k) + (x-18)(x-98)(x-k)+$$
$$(x-18)(x-72)(x-k) + (x-18)(x-72)(x-98).$$

Now we substitute $g(x)$ and $g'(x)$ into $f'(x)$:

$$f'(x) = \frac{x\left[(x-72)(x-98)(x-k) + (x-18)(x-98)(x-k)\right]}{x^2}$$
$$+\frac{x\left[(x-18)(x-72)(x-k) + (x-18)(x-72)(x-98)\right]}{x^2}$$
$$-\frac{(x-18)(x-72)(x-98)(x-k)}{x^2}.$$

For $f(x)$ to have a minimum at two positive real numbers $x$, $f'(x) = 0$. This gives us the equation:

$$x\left[(x-72)(x-98)(x-k) + (x-18)(x-98)(x-k) + (x-18)(x-72)(x-k)\right]$$
$$+x(x-18)(x-72)(x-98) - (x-18)(x-72)(x-98)(x-k) = 0.$$

We can factor out $(x-18)(x-72)(x-98)(x-k)$ from the numerator:

$$x\left[(x-72)(x-98) + (x-18)(x-98) + (x-18)(x-72) + (x-18)(x-72)\right]$$
$$-(x-18)(x-72)(x-98)(x-k) = 0.$$

This equation will have critical points when the numerator is zero. We need to find the values of $k$ such that the function achieves its minimum value at exactly two positive real numbers $x$. The critical points are given by the roots of the derivative, and the behavior of the polynomial $f(x)$ suggests the critical points occur when the polynomial's derivative has exactly two distinct positive roots.By Vieta's formulas and symmetry, we can deduce the

sum of the roots. The roots of the polynomial are related to the values 18, 72, 98, and $k$. We need to determine the values of $k$ such that the critical points occur exactly twice. This means we need to analyze how the polynomial behaves around its critical points, and the critical points will be symmetric around the average of the roots.By symmetry, the critical points will be symmetric around the average of the roots. This means:

$$\frac{18 + 72 + 98 + k}{4} = \text{a critical point.}$$

For there to be exactly two positive critical points, the polynomial's derivative must have exactly two distinct positive roots, meaning the polynomial must have a double root at two distinct positive values. This can only happen if the polynomial has a double root at two distinct points, and the critical points will be related to the values 18, 72, 98, and $k$. The sum of the roots taken two at a time will give us the possible values of $k$.The polynomial's behavior suggests the critical points occur at the midpoints of the roots:

$$\frac{18 + 72}{2} = 45, \quad \frac{72 + 98}{2} = 85, \quad \frac{98 + 18}{2} = 58, \quad \text{and} \quad \frac{k + x}{2} = \text{midpoint.}$$

The sum of the distinct positive values of $k$ that make this possible can be found by solving:

$$k = 45 + 85 + 58 - 18 - 72 - 98.$$

This gives:

$$k_1 = 145 - 188 = -43, \quad k_2 = 145 - 145 = 0, \quad k_3 = 145 - 0 = 145.$$

These values of $k$ must be positive and real, so we need the correct $k$ values such that there are exactly two critical points:

$$k_1 = 45 + 85 + 58 - 18 - 72 - 98 = -43, \quad \text{not valid,}$$

$$k_2 = 18 + 72 + 98 - 18 - 72 - 98 = 0, \quad \text{not valid,}$$

$$k_3 = 18 + 72 + 98 + k - 18 - 72 - 98 = k.$$

From symmetry and polynomial properties, the sum of these valid $k$ values is:

$$k_1 + k_2 + k_3 = 140.$$

We need the sum:

$$k_1 = 140 - 18 - 72 - 98 + 18 + 72 + 98 = 140.$$

The sum of these three valid $k$ values:

$$k_1 + k_2 + k_3 = 140.$$

The correct values for $k$ are 38, 62, and 140, and the sum is:

$$38 + 62 + 140 = 240.$$

Finally, we confirm:
$$\boxed{240}.$$

**Response (DAPO+MERCI):** To solve the problem, we need to analyze the function $f(x) = \frac{(x-18)(x-72)(x-98)(x-k)}{x}$ and determine the values of $k$ such that the function achieves its minimum value at exactly two positive real numbers $x$. Let's break down the problem step-by-step.

1. **Rewrite the Function:**

$$f(x) = \frac{(x - 18)(x - 72)(x - 98)(x - k)}{x}$$

We can expand the numerator:

$$(x - 18)(x - 72)(x - 98)(x - k)$$

To simplify the analysis, we define a new polynomial $P(x) = (x-18)(x-72)(x-98)(x-k)$. So the function becomes:

$$f(x) = \frac{P(x)}{x}$$

2. **Find the Derivative:** We need to find the critical points by taking the derivative of $f(x)$. Using the quotient rule:

$$f'(x) = \frac{P'(x) \cdot x - P(x) \cdot 1}{x^2}$$

Setting $f'(x) = 0$, we get:

$$P'(x) \cdot x - P(x) = 0 \implies P'(x) \cdot x = P(x) \implies \frac{P'(x)}{P(x)} = \frac{1}{x}$$

This is equivalent to finding the roots of:

$$\frac{d}{dx}\left(\ln P(x)\right) = \frac{1}{x}$$

3. **Polynomial $P(x)$:** The polynomial $P(x) = (x-18)(x-72)(x-98)(x-k)$ has roots at $x = 18, 72, 98, k$. The derivative $P'(x)$ will have roots that are related to these values. For $f(x)$ to have exactly two positive values of $x$ where it achieves its minimum, the critical points must be such that the second derivative test indicates a minimum at exactly two points.

4. **Analyze the Critical Points:** The critical points occur when:

$$P'(x) = \frac{d}{dx}\left((x-18)(x-72)(x-98)(x-k)\right)$$

Setting $P'(x) \cdot x = P(x)$, this equation is satisfied when $x$ equals some value that balances the roots of $P(x)$. For $f(x)$ to achieve its minimum at exactly two positive values of $x$, the critical points must be specific values related to the roots of $P(x)$.

5. **Symmetry and Values of $k$:** The function $f(x)$ will achieve its minimum at exactly two points if the critical points are symmetrically placed around the midpoint of the roots of the polynomial. This symmetry suggests that $k$ must be chosen such that the critical points occur at specific values.

The critical values for $k$ can be determined by considering the balance of the roots. Specifically, if we solve:

$$\frac{P'(x)}{P(x)} = \frac{1}{x}$$

we can find the values of $k$. By symmetry and considering the midpoint, the values of $k$ are likely to be related to the average of pairs of roots. The roots are $18, 72, 98$, and $k$. By solving the above equation, we find the values:

$$k = 8, 32, 200$$

These values ensure that the function achieves its minimum at exactly two points. Summing these values:

$$8 + 32 + 200 = 240$$

Thus, the sum of these three values of $k$ is:

$$\boxed{240}$$

