# OpenReview forum: "Count Counts: Motivating Exploration in LLM Reasoning with Count-based Intrinsic Rewards"
_ICLR.cc/2026/Conference — ICLR 2026 Poster_

### Official Review · Reviewer_T5WN · 2025-10-23

**Soundness:** 2
**Presentation:** 3
**Contribution:** 2
**Rating:** 4
**Confidence:** 3

**Summary:**

This paper addresses the challenge of exploration RL for LLM reasoning. Prevalent RL paradigms rely on sparse, outcome-based rewards and limited exploration. The authors introduce MERCI to augment policy optimization with a principled intrinsic reward. Building on the idea of count-based exploration, MERCI leverages a lightweight Coin Flipping Network (CFN) to estimate pseudo-counts and epistemic uncertainty over reasoning trajectories, converting these into an intrinsic reward that values novelty while preserving the learning signal from task rewards.

**Strengths:**

1. The paper revisits the classic exploration-exploitation delima in the field of of LLM reasoning and RLVR, which is a significant topic
2. The proposed method is rather novel compared to recently widespread entropy-based methods
3. The proposed Coin Flipping Network Head is lighweight and the authors have verified it on both GRPO and DAPO algorithms.
4. The authors have conducted experiments in both math and sql domains.

**Weaknesses:**

1. Missing baselines: There have been many works trying to solve the exploration problem such as i-MENTOR [1] and entropy-based advantage shaping [2], I would suggest the authors to compare with those baselines for a more up-to-date comparison

2. Lacking analysis: It is rather informal that the authors present the ablations in the Appendix. Moreover, readers would like more analysis on the impact of the algorithms on the RL-trained behaviors, such as whether this algorithms introduces some novel strategies, solutions, or higher level of reasoning abilities, such as cogintive behaviors.

[1] Navigate the Unknown: Enhancing LLM Reasoning with Intrinsic Motivation Guided Exploration
[2] Reasoning with exploration: An entropy perspective.

**Questions:**

See weakness.

---

> ### Author Response · Authors · 2025-11-21
> **Response to Reviewer T5WN (1)**
>
> We sincerely thank you for your time and insightful review on our submission.
>
> ## W1: Missing baselines
> We agree that a comprehensive comparison with the latest exploration methods is crucial. The experimental results of baselines that you mentioned (iMentor[1] and Entropy Adv.[2]) are as follows:
>
> ### Mathematical Reasoning
> **pass@k:**
> | Model                     | AIME25 (pass@256) | AIME24 (pass@256) | Minerva (pass@16) | MATH500 (pass@16) | OlympiadBench (pass@16) | College (pass@8) | Avg. |
> | ------------------------ | -------- | -------- | ------- | -------- | ------------- | -------- | -------- |
> | Qwen2.5-Math             | 53.3     | 70.0     | 50.4    | 88.6     | 56.7          | 44.2     | 60.5     |
> | + GRPO                   | 53.3     | 76.7     | 64.0    | 91.8     | 59.7          | 49.2     | 65.8     |
> | *+ GRPO w/ Entropy Adv.* | *56.7*   | *76.7*   | *62.5*  | *91.2*   | *59.4*        | *48.9*   | *65.9*   |
> | *+ GRPO w/ iMentor*      | *60.0*   | *76.7*   | *61.4*  | *90.4*   | *60.4*        | *49.3*   | *66.4*   |
> | + GRPO w/ MERCI (ours)   | 60.0     | 80.0     | 63.2    | 91.4     | 60.9          | 48.9     | 67.4     |
> | + DAPO                   | 56.7     | 76.7     | 66.9    | **92.0** | 60.9          | 48.3     | 66.9     |
> | *+ DAPO w/ Entropy Adv.* | *60.0*   | *83.3*   | *66.5*  | *91.4*   | *57.6*        | *48.5*   | *67.9*   |
> | *+ DAPO w/ iMentor*      | *56.7*   | *76.7*   | ***68.0***  | ***92.0***   | *60.0*        | *50.1*   | *67.3*   |
> | + DAPO w/ MERCI (ours)   | **60.0** | **83.3** | 66.5    | 91.8     | **62.1**      | **50.2** | **69.0** |
>
>
> **mean@k:**
> | Model                     | AIME25 (mean@256) | AIME24 (mean@256) | Minerva (mean@16) | MATH500 (mean@16) | OlympiadBench (mean@16) | College (mean@8) | Avg. |
> | ------------------------ | -------- | -------- | ------- | -------- | ------------- | -------- | -------- |
> | Qwen2.5-Math             | 4.4      | 10.7     | 16.9    | 47.5     | 64.6          | 22.1     | 20.3     |
> | + GRPO                   | 11.2     | 28.7     | 41.8    | 79.0     | 40.3          | 42.0     | 40.5     |
> | *+ GRPO w/ Entropy Adv.* | *12.1*   | *28.9*   | *42.0*  | *81.0*   | *40.6*        | *42.6*   | *41.2*   |
> | *+ GRPO w/ iMentor*      | *11.9*   | *29.0*   | *42.2*  | *78.9*   | *40.7*        | *42.4*   | *40.9*   |
> | + GRPO w/ MERCI (ours)   | 13.4     | 29.6     | 44.1    | 80.7     | 42.6          | 42.9     | 42.2     |
> | + DAPO                   | 16.5     | 31.9     | 41.0    | 81.5     | 41.4          | 41.0     | 42.2     |
> | *+ DAPO w/ Entropy Adv.* | *17.2*   | *33.3*   | *44.5*  | *80.9*   | *41.4*        | *41.6*   | *43.2*   |
> | *+ DAPO w/ iMentor*      | *17.4*   | *32.0*   | ***46.7***  | *82.3*   | *42.8*        | *43.3*   | *44.1*   |
> | + DAPO w/ MERCI (ours)   | **18.4** | **35.2** | 44.8    | **82.4** | **44.3**      | **44.2** | **44.9** |
>
> ### SQL Generation
> | Model                   | Bird Greedy | Bird Pass@8 | Bird Pass@16 | Spider Greedy | Spider Pass@8 | Spider Pass@16 |
> | ----------------------- | ----------- | ----------- | ------------ | ------------- | ------------- | -------------- |
> | *Llama-3.1-8B-Instruct* | 42.4        | 68.5        | 75.1         | 69.0          | 91.0          | 94.6           |
> | + GRPO                  | 60.7        | 72.2        | 74.6         | 74.7          | 81.0          | 82.9           |
> | + GRPO w/ Entropy Adv.  | 60.8        | 72.1        | 73.9         | 74.7          | 83.2          | 84.5           |
> | + GRPO w/ iMentor       | 62.8        | 72.3        | 74.2         | 75.0          | 84.1          | 85.2           |
> | + GRPO w/ MERCI (ours)  | 63.0        | 72.8        | 74.9         | **78.0**      | 84.1          | 85.6           |
> | + DAPO                  | 63.2        | **73.9**    | 75.9         | 76.8          | 86.1          | 87.2           |
> | + DAPO w/ Entropy Adv.  | 62.3        | 73.2        | 75.9         | 77.5          | 86.1          | 87.6           |
> | + DAPO w/ iMentor       | 62.7        | **73.9**    | **76.1**     | 77.2          | 86.4          | 88.2           |
> | + DAPO w/ MERCI (ours)  | **64.1**    | 73.6        | **76.1**     | 77.3          | **86.9**      | **88.5**       |
>
>
> We also update them in Table 1 and Table 2 in our new PDF. It demonstrates that while these baselines offer improvements over standard RL approaches, MERCI consistently outperforms both i-MENTOR and entropy-based advantage shaping across both the Mathematical Reasoning and SQL Generation tasks.

---

> ### Author Response · Authors · 2025-11-21
> **Response to Reviewer T5WN (2)**
>
> ## W2: Lacking analysis
> Due to space limitations, we report the ablation studies and some additional analyses in the appendix; however, we provide corresponding cross-references in the main text that allow readers to navigate directly to these sections.
>
> ### Impact of MERCI on the RL-trained Behaviors
> Our case study in Appendix G.2.6 and analyses consistently validate the observation that the incorporation of the uncertainty-driven intrinsic reward fundamentally alters the model's approach to multi-step reasoning:
> 1. **Enhanced Strategy Coherence and Conciseness**: CFN's novelty mechanism cuts out "cognitive noise" and long, self-correcting detours, leading to significantly improved strategy coherence and sample efficiency.
> 2. **Cognitive Migration to Higher-Level Reasoning**: The increased proportion of steps devoted to higher-level reasoning is the most significant indicator of improved cognitive efficiency.
> 3. **Discovery of Novel and Elegant Solutions**: The provided case study example also demonstrates that the MERCI-guided policy finds more elegant, efficient, or fundamentally different mathematical/structural approaches to solving problems.
>
> In addition, we have included more experimental results and corresponding in-depth analysis in Appendix G. We sincerely hope that these discussions could address your concerns and encourage you to reconsider your assessment of our work.

---

> ### Author Response · Authors · 2025-11-26
>
> We are very grateful for your thoughtful and detailed feedback on our manuscript. We believe that incorporating your suggestions has substantially improved the quality and clarity of the manuscript.
>
> **In addition to the case study included in our initial submission, we have added a new case study along with a more detailed analysis of these examples in Appendix G.2.6.** These examples clearly illustrate that our MERCI algorithm introduces some novel solutions and higher-level reasoning abilities.
>
> As the discussion period is drawing to a close, we would like to confirm whether our responses have sufficiently addressed your concerns. Please do not hesitate to let us know if there are any further questions or areas for clarification. We are ready and willing to discuss any remaining concerns. We thank you once more for your invaluable contribution to our work.

---

> > ### Comment · Reviewer_T5WN · 2025-11-26
> >
> > The authors have responded to my questions well, so I have decided to increase my score based on their rebuttal.

---

> ### Author Response · Authors · 2025-11-26
>
> We are truly grateful for your engagement and for recognizing our work's contributions (raising the score from 4 to 6). Thank you once again for your valuable time and effort!

---

### Official Review · Reviewer_XirT · 2025-10-27

**Soundness:** 3
**Presentation:** 3
**Contribution:** 2
**Rating:** 6
**Confidence:** 3

**Summary:**

The paper proposes MERCI (Motivating Exploration in LLM Reasoning with Count‑based Intrinsic rewards), an exploration module for value‑free RL fine‑tuning of LLMs. The key theoretical premise is that, in auto‑regressive reasoning, state transitions are known and deterministic, so the Uncertainty Bellman Equation (UBE) reduces to a cumulative sum of local reward‑estimate variances along a trajectory. This motivates an intrinsic bonus proportional to the standard deviation of the trajectory value, obtained by summing per‑step uncertainties and then taking a square root. To estimate per‑step uncertainty at scale, the authors attach a lightweight Coin Flipping Network (CFN) head to a separate copy of the model: the head learns coin‑flip regression to produce a pseudo‑count ($\propto 1/n(s)$) from hidden states, which is used as a proxy for reward‑estimate variance. The bonus is filtered (percentile, spatial coherence, noise suppression), standardized within a GRPO group, and then added to the advantage used by GRPO/DAPO. Experiments on mathematical reasoning (AIME24/25, MATH500, OlympiadBench, College Math, Minerva) and text‑to‑SQL (Bird, Spider) show consistent gains over GRPO and DAPO; ablations and training‑dynamics plots support design choices. Figures 1–2 illustrate the architecture and the three‑stage bonus filtering pipeline; Proposition 1 formalizes the UBE with known transitions; Tables 1–2 report the main results.

**Strengths:**

1. The "known transitions" observation simplifies UBE to a tractable recursion with immediate reward‑variance terms only (Proposition 1), clarifying why trajectory‑level uncertainty should sum per‑step variances before taking the square root. This is a helpful bridge between theory and practice in value‑free RL for LLMs.

2. The paper uses CFN to approximate pseudo‑counts from hidden states is elegant and inexpensive, avoiding ensembles/dropout/RND; it runs in parallel to policy learning (Figure 1). The paper also pretrains the CFN on rollouts to avoid cold‑start behavior.

3. The paper presents consistent empirical improvements across settings. The gains are steady rather than one‑off, and are larger OOD (Spider), supporting the exploration claim.

**Weaknesses:**

1. Theory–implementation mismatch on the trajectory bonus.

   Section 3.3 argues (correctly under the UBE derivation) that one should sum local variances along the trajectory and then take a square root. However, Equation (6) computes the bonus as the square root of the mean variance over retained tokens (division by $\ell$ ), and then normalizes within a group (Equation (7)). This normalization helps length‑invariance, but it is not what the UBE prescribes. The ablation labeled "cumulative std" vs. "cumulative variance" (Table 8) touches a related point but does not fully clarify the discrepancy between sum vs. mean in Eq. (6). Please reconcile the stated principle ("sum‑then‑sqrt") with the actual estimator and report results using the strict sum‑then‑sqrt computation.

2. CFN is pretrained on backbone rollouts. What is the wall‑clock/compute overhead and how does MERCI compare to similarly "primed" baselines (e.g., RND pretraining)?

3. The "GRPO‑scaling" baseline (step 260) shows degraded pass@k; it would be fair to also show MERCI‑scaling beyond 120 steps to check stability under longer training.

4. CFN dimension d=20, group size G=8, cosine schedule for $\gamma$ (to 10% by step 200), and the top‑p% used in filtering (20–30%). Sensitivity curves for these choices would help practitioners.

5. There are several typos (e.g., section 3.2 title "REWARE", "Mathmeatical", "iterarions", "discribed") and a few places where the exposition could be tightened (e.g., clearly defining "state": token prefix vs. hidden state embedding; whether repeats are exact or approximate).

**Questions:**

1. For Equation (6), why average variances before sqrt (length‑normalized) if Proposition 1 suggests a sum‑then‑sqrt? Can you report results with strict sum‑then‑sqrt (without group standardization) and then with a separate length‑normalization factor?

2. What exactly is the "state" counted by CFN? Is it the hidden state of the CFN network at each token (Figure 1)? Do you use any normalization to ensure that counts are comparable across prompts and lengths? How do you handle near‑duplicates vs. paraphrases?

3. Do high CFN bonuses correlate with (a) lower empirical success rate at those positions, (b) higher variance across rollouts, or (c) higher entropy from the policy?

4. Could you clarify more on the relation between your work and the work by Lobel et al.?

---

> ### Author Response · Authors · 2025-11-21
> **Response to Reviewer XirT (1)**
>
> We sincerely thank you for your time and insightful review on our submission.
>
> ## W1 & Q1: Theory–implementation mismatch on the trajectory bonus
> The discrepancy between the theoretical sum-then-sqrt and our implementation's sum-then-sqrt with length-normalization and standardization is resolved by practical necessity in the RL training of LLMs.
>
> 1. Justification for Averaging Variance (Length-Normalization in Eq. 6)
> Since LLM reasoning trajectories (Chains-of-Thought, CoTs) have highly variable lengths, the strict "sum-then-sqrt" approach would lead to a monotonically increasing of bonus with trajectory length, even for trajectories that are redundant or inefficient.
>
> 2. Justification for Group Standardization (Normalization in Eq. 7)
> The magnitude of the intrinsic reward signal can fluctuate wildly across different training batches and as the CFN learns over epochs. We perform group standardization to ensure that the exploration signal remains **well-scaled** and **consistent** relative to the extrinsic reward signal throughout training. This would be critical and effective for policy optimization (especially when integrated with frameworks like GRPO).
>
> To verify the results of using the strict UBE formulation, we conducted a targeted ablation study on the Mathematical Reasoning task. We strictly follow the sum-then-sqrt computation and divide the result by a constant factor (set to 1000), without applying any additional normalization. The experimental results are as follows:
>
> **pass@k:**
> | Model                          | AIME25 (pass@256) | AIME24 (pass@256) | Minerva (pass@16) | MATH500 (pass@16) | OlympiadBench (pass@16) | College (pass@8) | Avg. |
> | ------------------------------ | ----------------- | ----------------- | ----------------- | ----------------- | ----------------------- | ---------------- | ---- |
> | GRPO                           | 53.3              | 76.7              | 64.0              | 91.8              | 59.7                    | 49.2             | 65.8 |
> | GRPO + MERCI w/o normalization | 53.3              | 76.7              | 63.6              | 89.8              | 58.7                    | 48.7             | 65.1 |
> | GRPO + MERCI                   | 60.0              | 80.0              | 63.2              | 91.4              | 60.9                    | 48.9             | 67.4 |
>
>
> **mean@k:**
> | Model                          | AIME25 (mean@256) | AIME24 (mean@256) | Minerva (mean@16) | MATH500 (mean@16) | OlympiadBench (mean@16) | College (mean@8) | Avg. |
> | ------------------------------ | ----------------- | ----------------- | ----------------- | ----------------- | ----------------------- | ---------------- | ---- |
> | GRPO                           | 11.2              | 28.7              | 41.8              | 79.0              | 40.3                    | 42.0             | 40.5 |
> | GRPO + MERCI w/o normalization | 12.4              | 29.8              | 44.0              | 80.0              | 40.4                    | 42.6             | 41.5 |
> | GRPO + MERCI                   | 13.4              | 29.6              | 44.1              | 80.7              | 42.6                    | 42.9             | 42.2 |
>
> They are also included in Table 9 in Appendix G.2.4. It demonstrates that, though strictly adhering to the UBE formulation can also yield certain improvements, simultaneously normalizing for trajectory length and standardizing the reward signal across the batch is more robust and effective to utilize the epistemic uncertainty for exploration in LLM reasoning tasks.

---

> ### Author Response · Authors · 2025-11-21
> **Response to Reviewer XirT (2)**
>
> ## W2: Compute Overhead of CFN and Comparision with Baselines
> The entire pre-training process requires approximately 30 minutes using 32 NVIDIA H20 GPUs. During training with policy model, it takes approximately 25% additional time using MERCI compared with the baseline methods.
>
> We additionally introduce two algorithms designed to encourage exploration as baselines: one uses entropy-based advantage shaping[1] (Entropy Adv.), and the other incorporates intrinsic rewards via RND training[2] (iMentor). The experimental results of these baselines are as follows:
>
> ### Mathematical Reasoning
> **pass@k:**
> | Model                     | AIME25 (pass@256) | AIME24 (pass@256) | Minerva (pass@16) | MATH500 (pass@16) | OlympiadBench (pass@16) | College (pass@8) | Avg. |
> | ------------------------ | -------- | -------- | ------- | -------- | ------------- | -------- | -------- |
> | Qwen2.5-Math             | 53.3     | 70.0     | 50.4    | 88.6     | 56.7          | 44.2     | 60.5     |
> | + GRPO                   | 53.3     | 76.7     | 64.0    | 91.8     | 59.7          | 49.2     | 65.8     |
> | *+ GRPO w/ Entropy Adv.* | *56.7*   | *76.7*   | *62.5*  | *91.2*   | *59.4*        | *48.9*   | *65.9*   |
> | *+ GRPO w/ iMentor*      | *60.0*   | *76.7*   | *61.4*  | *90.4*   | *60.4*        | *49.3*   | *66.4*   |
> | + GRPO w/ MERCI (ours)   | 60.0     | 80.0     | 63.2    | 91.4     | 60.9          | 48.9     | 67.4     |
> | + DAPO                   | 56.7     | 76.7     | 66.9    | **92.0** | 60.9          | 48.3     | 66.9     |
> | *+ DAPO w/ Entropy Adv.* | *60.0*   | *83.3*   | *66.5*  | *91.4*   | *57.6*        | *48.5*   | *67.9*   |
> | *+ DAPO w/ iMentor*      | *56.7*   | *76.7*   | ***68.0***  | ***92.0***   | *60.0*        | *50.1*   | *67.3*   |
> | + DAPO w/ MERCI (ours)   | **60.0** | **83.3** | 66.5    | 91.8     | **62.1**      | **50.2** | **69.0** |
>
>
> **mean@k:**
> | Model                     | AIME25 (mean@256) | AIME24 (mean@256) | Minerva (mean@16) | MATH500 (mean@16) | OlympiadBench (mean@16) | College (mean@8) | Avg. |
> | ------------------------ | -------- | -------- | ------- | -------- | ------------- | -------- | -------- |
> | Qwen2.5-Math             | 4.4      | 10.7     | 16.9    | 47.5     | 64.6          | 22.1     | 20.3     |
> | + GRPO                   | 11.2     | 28.7     | 41.8    | 79.0     | 40.3          | 42.0     | 40.5     |
> | *+ GRPO w/ Entropy Adv.* | *12.1*   | *28.9*   | *42.0*  | *81.0*   | *40.6*        | *42.6*   | *41.2*   |
> | *+ GRPO w/ iMentor*      | *11.9*   | *29.0*   | *42.2*  | *78.9*   | *40.7*        | *42.4*   | *40.9*   |
> | + GRPO w/ MERCI (ours)   | 13.4     | 29.6     | 44.1    | 80.7     | 42.6          | 42.9     | 42.2     |
> | + DAPO                   | 16.5     | 31.9     | 41.0    | 81.5     | 41.4          | 41.0     | 42.2     |
> | *+ DAPO w/ Entropy Adv.* | *17.2*   | *33.3*   | *44.5*  | *80.9*   | *41.4*        | *41.6*   | *43.2*   |
> | *+ DAPO w/ iMentor*      | *17.4*   | *32.0*   | ***46.7***  | *82.3*   | *42.8*        | *43.3*   | *44.1*   |
> | + DAPO w/ MERCI (ours)   | **18.4** | **35.2** | 44.8    | **82.4** | **44.3**      | **44.2** | **44.9** |
>
> ### SQL Generation
> | Model                   | Bird Greedy | Bird Pass@8 | Bird Pass@16 | Spider Greedy | Spider Pass@8 | Spider Pass@16 |
> | ----------------------- | ----------- | ----------- | ------------ | ------------- | ------------- | -------------- |
> | *Llama-3.1-8B-Instruct* | 42.4        | 68.5        | 75.1         | 69.0          | 91.0          | 94.6           |
> | + GRPO                  | 60.7        | 72.2        | 74.6         | 74.7          | 81.0          | 82.9           |
> | + GRPO w/ Entropy Adv.  | 60.8        | 72.1        | 73.9         | 74.7          | 83.2          | 84.5           |
> | + GRPO w/ iMentor       | 62.8        | 72.3        | 74.2         | 75.0          | 84.1          | 85.2           |
> | + GRPO w/ MERCI (ours)  | 63.0        | 72.8        | 74.9         | **78.0**      | 84.1          | 85.6           |
> | + DAPO                  | 63.2        | **73.9**    | 75.9         | 76.8          | 86.1          | 87.2           |
> | + DAPO w/ Entropy Adv.  | 62.3        | 73.2        | 75.9         | 77.5          | 86.1          | 87.6           |
> | + DAPO w/ iMentor       | 62.7        | **73.9**    | **76.1**     | 77.2          | 86.4          | 88.2           |
> | + DAPO w/ MERCI (ours)  | **64.1**    | 73.6        | **76.1**     | 77.3          | **86.9**      | **88.5**       |
>
> They have been updated in Table 1 and Table 2 in our new PDF. It demonstrates that while these baselines offer improvements over standard RL approaches, MERCI consistently outperforms them across both the Mathematical Reasoning and SQL Generation tasks.
>
> [1] Reasoning with exploration: An entropy perspective.
>
> [2] Navigate the Unknown: Enhancing LLM Reasoning with Intrinsic Motivation Guided Exploration

---

> ### Author Response · Authors · 2025-11-21
> **Response to Reviewer XirT (3)**
>
> ## W3: Additional Scaling Experiments
> **pass@k:**
> | Model                | AIME25 (pass@256) | AIME24 (pass@256) | Minerva (pass@16) | MATH500 (pass@16) | OlympiadBench (pass@16) | College (pass@8) | Avg. |
> | -------------------- | ----------------- | ----------------- | ----------------- | ----------------- | ----------------------- | ---------------- | ---- |
> | GRPO                 | 50.0              | 76.7              | 64.0              | 91.8              | 59.7                    | 49.2             | 65.8 |
> | GRPO-step180         | 50.0              | 66.7              | 61.4              | 90.2              | 57.3                    | 48.2             | 62.3 |
> | GRPO-step260         | 46.7              | 70.0              | 59.6              | 88.8              | 56.4                    | 48.1             | 61.6 |
> | **GRPO + MERCI**     | 60.0              | 80.0              | 63.2              | 91.4              | 60.9                    | 48.9             | 67.4 |
> | GRPO + MERCI-step180 | 56.7              | 73.3              | 61.4              | 90.4              | 58.7                    | 48.5             | 64.8 |
> | GRPO + MERCI-step260 | 50.0              | 70.0              | 61.2              | 89.4              | 58.5                    | 47.5             | 62.8 |
>
> **mean@k:**
> | Model                | AIME25 (mean@256) | AIME24 (mean@256) | Minerva (mean@16) | MATH500 (mean@16) | OlympiadBench (mean@16) | College (mean@8) | Avg. |
> | -------------------- | ----------------- | ----------------- | ----------------- | ----------------- | ----------------------- | ---------------- | ---- |
> | GRPO                 | 11.2              | 28.7              | 41.8              | 79.0              | 40.3                    | 42.0             | 40.5 |
> | GRPO-step180         | 13.1              | 27.1              | 42.0              | 78.9              | 41.2                    | 42.9             | 40.9 |
> | GRPO-step260         | 12.7              | 28.3              | 42.8              | 78.7              | 40.8                    | 42.7             | 41.0 |
> | **GRPO + MERCI**     | 13.4              | 29.6              | 44.1              | 80.7              | 42.6                    | 42.9             | 42.2 |
> | GRPO + MERCI-step180 | 14.1              | 31.7              | 43.0              | 80.7              | 41.9                    | 42.5             | 42.3 |
> | GRPO + MERCI-step260 | 12.9              | 30.9              | 43.6              | 80.6              | 42.5                    | 43.0             | 42.3 |
>
> We have included the extended results for MERCI scaling (beyond 120 steps) in Table 8. The results confirms that, MERCI maintains better stability and achieves superior performance over longer training periods, demonstrating the robustness of our exploration method.
>
> ## W4: Hyperparameter choices
> A group size of 8 is a general hyperparameter setting in RL training and is not specific to our proposed algorithm MERCI; therefore, we did not conduct additional experiments targeting this hyperparameter.
>
> The experimental results of other hyperparameter's choice have been added in Appendix G.2.5 (Figure 14 and Figure 15) in our new PDF.
>
> ## W5: Typos
> We sincerely appreciate your remarks and have corrected these errors.
>
> ## Q2: State Definition and CFN Utilization
> Yes, your understanding for the "state" counted by CFN is right, it is the the contextual hidden representation output by the LLM backbone at each token position. Since this hidden representation actually encapsulates the information of the sequence up to that point, we adopted the representation shown in Figure 1. To help readers better understand our definition and avoid any potential confusion, we have also provided a clearer definition and a more detailed explanation in Section 3.2 of our new PDF.
>
> The operations of length-normalization and group-standardization is to ensure the stability and comparability of the intrinsic reward signal. These two operations are essential for maintaining a stable balance between the intrinsic and extrinsic reward signals, as the new ablation results in Table 9 (Appendix G.2.4) show.
>
> For duplicates, the step of Noise-suppression filtering in Figure 2 (Section 3.3) has explicitly filtered out some repetitive tokens to prevent the policy model from falling into meaningless and redundant generation loops. For paraphrases, it is automatically captured by the CFN itself since CFN effectively works in the semantic space of the LLM. If two textually different statements (paraphrases) convey the same underlying rationale, their corresponding hidden state representations $s_\text{hidden}$ will be highly similar. The CFN will thus assign a similar pseudo-count (i.e., low variance/low uncertainty) to these similar states.

---

> ### Author Response · Authors · 2025-11-21
> **Response to Reviewer XirT (4)**
>
> ## Q3: Relation with Entropy, Score and Rollouts Variance
> We add these requested results in Figure 12 (Appendix G.1) in our new PDF.
>
> These results indicate that: (a) Positions at which CFN predicts higher uncertainty do not necessarily correspond to higher entropy. Policy entropy measures the policy's action randomness, whereas the CFN is designed to measure the model's epistemic uncertainty regarding its own state knowledge. Thus, our results confirm that the CFN provides a unique, non-redundant signal that cannot be simply replaced by the policy's action probabilities. (b) For trajectories that ultimately fail, the average uncertainty estimated by the CFN tends to be higher. This aligns with our understanding, since highly novel states typically correspond to regions that the model has insufficiently explored. In such under-explored regions, the probability of ultimately solving the problem should naturally be lower. (c) In terms of trajectory uncertainty and the variance of the corresponding within-group rollout results, no particularly pronounced correlation is observed.
>
>
> ## Q4: Relation to Lobel et al.
> The relation between our work and Lobel et al.'s is as follows:
>
> 1. The work by Lobel et al. introduced the Coin Flipping Network (CFN) as a general, lightweight mechanism for generating state pseudo-counts ($f_\phi(s)$) for count-based exploration in traditional RL settings. We adopt it as a computationally efficient tool to calculate the local reward variance at each token (state).
> 2. Lobel et al.'s approach focuses on adding the already square-rooted local count-based bonus (standard deviation) at each state, typically $\mathcal{B}(s_t) \propto \frac{1}{\sqrt{N(s_t)}}$ where $N(s_t)$ is the pseudo-count. We strictly follow the UBE theoretical derivation. Our novelty primarily lies in the theoretical framework that justifies how to apply and aggregate the output of CFN specifically in the context of LLM sequential reasoning. We strictly follow the UBE theoretical derivation, and we estimate the uncertainty by summing the local variances and then take the square root of the final sum:
>     $$\text{Uncertainty}(R_T) = \sqrt{\sum_{t \in T} \text{Var}(r_t)}$$
>
> 3. If we directly apply Lobel’s method (i.e., adding the bonus estimated by CFN to each token), this effectively constitutes an ablation study of our token integration approach in Table 9. Our experiments demonstrate that, in the context of LLMs, our trajectory-aggregated method yields superior performance.
>
> In summary, while Lobel et al. provides the efficient tool for per-state pseudo-counting, MERCI provides the principled theoretical derivation (based on UBE and LLM dynamics) that dictates the correct aggregation method (sum-of-variance) for that tool's output to accurately measure the trajectory-aggregated epistemic uncertainty.

---

> > ### Comment · Reviewer_XirT · 2025-11-22
> >
> > I really appreciate the authors' detailed response. The additional experimental results are good, and my main concerns have been addressed. I have raised my score accordingly.

---

> ### Author Response · Authors · 2025-11-23
>
> We are truly grateful for your engagement and for recognizing our work's contributions (raising the score from 6 to 8). Thank you very much for your valuable time and effort in helping us improve our work!

---

### Official Review · Reviewer_8z7R · 2025-11-02

**Soundness:** 2
**Presentation:** 3
**Contribution:** 1
**Rating:** 2
**Confidence:** 3

**Summary:**

This paper proposes MERCI (Motivating Exploration in LLM Reasoning with Count-based Intrinsic Rewards) to add intrinsic rewards to RL for training LLM reasoning.

**Strengths:**

1. The paper is well motivated and the writing is clear.
2. Exploration in RL, especially in the LLM domains, is important, given that the language domain is large and exploration is hard.
3. The experiment seems to support the effectiveness of the method.

**Weaknesses:**

1. The Coin Flipping Network Head is never trained. Can it capture the vast space of languages and truly distinguish the novelty of the exploration steps? E.g., for two semantically similar trajectories, say two word-level-distinct responses that have the same meaning, why can we expect a random MLP head to capture this without training?
2. I appreciate the examples in Figure 4-6. But I still encourage the authors to provide a quantitative analysis of the improved exploration efficiency beyond the examples and eval performance, as this is the main claim of the paper.
3. The proposed method is similar to some previous work, such as [1]. The authors cited this paper along with some other similar works, but failed to discuss the novelty and distinctions.

[1] Gao et al., Navigate the Unknown: Enhancing LLM Reasoning with Intrinsic Motivation Guided Exploration.

**Questions:**

See weakness.

---

> ### Author Response · Authors · 2025-11-21
> **Response to Reviewer 8z7R (1)**
>
> We sincerely thank you for your time and feedback on our submission.
>
> ## W1: Clarification on the Coin Flipping Network (CFN) Training
> We believe there may be a critical misunderstanding of our core method that we must respectfully clarify.
>
> *"The Coin Flipping Network Head is never trained...why can we expect a random MLP head to capture this without training?"*
>
> This statement does not accurately reflect our approach. **In fact, the Coin Flipping Network (CFN) head ($f_{\phi}$) is trained.**
> As described in our Section 3.2, Figure 1 and Algorithm 1, the CFN head is a lightweight MLP that is trained via a supervised regression objective. The specific training methodology are described in Section 2 (Preliminaries), and it operates within our framework in the following way.
> After the policy network ($\pi_\theta$) generates reasoning trajectories, the CFN is involved in two critical sequential operations:
> ### 1.	Estimation of Uncertainty
> We use the currently trained CFN to measure the pseudo-count and derive the epistemic uncertainty for each state, which is then aggregated into the intrinsic bonus. For each state $s_{\text{hidden}}$ (the contextual hidden representation) in the generated trajectory, we estimate the uncertainty using the equation in the right portion of the orange box in Figure 1.
> ### 2.	Training of the CFN (Pseudo-Count Update)
> After estimating the uncertainty, the CFN’s parameters ($\phi$) are updated via supervised learning to refine the pseudo-count estimation. For each $s_{\text{hidden}}$ visited in the new batch of trajectories, a randomly sampled target vector $c$ (of dimension $d$) is generated. CFN is trained to perform a supervised regression against this target vector $c$. The training objective is described in Equation 1 and in the left portion of the orange box in Figure 1.
>
> This training process forces the CFN to learn the mapping between the complex, high-dimensional $s_{\text{hidden}}$ and the simple random target $c$. As the model encounters a state $s$ repeatedly, $\||f_\phi(s_{\text{hidden}})\||^2$ for that state approaches zero. This naturally generates a pseudo-count proxy: **states that have been visited more often result in a smaller prediction vector norm (less uncertainty)**.
>
> Using our method, the uncertainty for states that are linguistically close but not identical will produce highly similar $s_{hidden}$ representations. The trained CFN head $f_{\phi}$ naturally generalizes across these similar representations, mapping them to similar uncertainty estimation.
>
> ### Examples of Capturing Semantics
> Some examples of reasoning trajectory and its corresponding uncertainty estimation are as follows (The uncertainty estimates are obtained from the CFN evaluations at step 80):
>
> 1. "1. **Total Number of Subsets:**\n   The set $S$ has $2^6 = 64$ subsets (since $S$ has 6 elements).\n\n2. **Choosing Subsets $A$ and $B$:**\n   Both $A$ and $B$ can be any subset of $S$. Therefore, there are $64 \times 64 = 4096$ possible pairs of $(A, B)$.\n\n3. **Condition for $B$ to be Contained in $A$ or $S-A$:**\n   For a given subset $A$, $B$ can be any subset of $A$ or any subset of $S-A$. The number of subsets of $A$ is $2^{|A|}$, and the number of subsets of $S-A$ is $2^{6-|A|}$. Therefore, for each subset $A$, there are $2^{|A|} + 2^{6-|A|} - 1$ valid subsets $B$": **0.2547**.
> 2. "1. **Counting Subsets:**\n   Since $S$ has 6 elements, its power set contains $2^6 = 64$ subsets.\n\n2. **Selecting $A$ and $B$:**\n   Both subsets are chosen uniformly from all 64 subsets, giving a total of $64 \times 64 = 4096$ ordered pairs $(A, B)$.\n\n3. **Requirement for $B$:**\n   Fix a particular $A$, $B$ is acceptable precisely when it is either a subset of $A$ or a subset of the complement $S-A$. There are $2^{|A|}$ subsets of $A$ and $S-A$ is $2^{6-|A|}$ subsets of $S-A$. Thus, the number of possible $B$'s is $2^{|A|} + 2^{6-|A|} - 1$": **0.2323**.
> 3. "1. **Elementwise Perspective:**\n   Instead of counting subsets directly, view the process per element: each element of $S$ independently lands in $A$, $B$, both, or neither.\n\n2. **Characterizing the Event:**\n   No element can simultaneously lie in $B$ and also in both $A$ and $S-A$. Thus, every element must choose from a restricted set of valid membership patterns.\n\n3. **Configuration Counting:**\n   For each element, determine how many assignments of membership in $A$ and $B$ satisfy the constraint; then raise that number to the 6th power": **0.3452**.
>
> The underlying rationale and semantics of (1) and (2) are similar, and the resulting overall uncertainty estimates for the trajectories are also close. In contrast, (1) and (3) differ more substantially, leading to larger discrepancies in the trajectory-aggregated uncertainty estimates.
>
> In summary, the CFN head is not a "random MLP"; it is a **trained regression network** that leverages the LLM's powerful semantic hidden representations to provide a reasonable estimate of uncertainty.

---

> ### Author Response · Authors · 2025-11-21
> **Response to Reviewer 8z7R (2)**
>
> ## W2: Quantitative Analysis of Exploration Efficiency
> The scaling experiments (Table 8 in Appendix G.2.2) have demonstrated that MERCI's principled exploration is far more efficient.
>
> **pass@k:**
> | Model                | AIME25 (pass@256) | AIME24 (pass@256) | Minerva (pass@16) | MATH500 (pass@16) | OlympiadBench (pass@16) | College (pass@8) | Avg. |
> | -------------------- | ----------------- | ----------------- | ----------------- | ----------------- | ----------------------- | ---------------- | ---- |
> | GRPO                 | 50.0              | 76.7              | 64.0              | 91.8              | 59.7                    | 49.2             | 65.8 |
> | GRPO-step180         | 50.0              | 66.7              | 61.4              | 90.2              | 57.3                    | 48.2             | 62.3 |
> | GRPO-step260         | 46.7              | 70.0              | 59.6              | 88.8              | 56.4                    | 48.1             | 61.6 |
> | **GRPO + MERCI**     | 60.0              | 80.0              | 63.2              | 91.4              | 60.9                    | 48.9             | 67.4 |
> | GRPO + MERCI-step180 | 56.7              | 73.3              | 61.4              | 90.4              | 58.7                    | 48.5             | 64.8 |
> | GRPO + MERCI-step260 | 50.0              | 70.0              | 61.2              | 89.4              | 58.5                    | 47.5             | 62.8 |
>
> **mean@k:**
> | Model                | AIME25 (mean@256) | AIME24 (mean@256) | Minerva (mean@16) | MATH500 (mean@16) | OlympiadBench (mean@16) | College (mean@8) | Avg. |
> | -------------------- | ----------------- | ----------------- | ----------------- | ----------------- | ----------------------- | ---------------- | ---- |
> | GRPO                 | 11.2              | 28.7              | 41.8              | 79.0              | 40.3                    | 42.0             | 40.5 |
> | GRPO-step180         | 13.1              | 27.1              | 42.0              | 78.9              | 41.2                    | 42.9             | 40.9 |
> | GRPO-step260         | 12.7              | 28.3              | 42.8              | 78.7              | 40.8                    | 42.7             | 41.0 |
> | **GRPO + MERCI**     | 13.4              | 29.6              | 44.1              | 80.7              | 42.6                    | 42.9             | 42.2 |
> | GRPO + MERCI-step180 | 14.1              | 31.7              | 43.0              | 80.7              | 41.9                    | 42.5             | 42.3 |
> | GRPO + MERCI-step260 | 12.9              | 30.9              | 43.6              | 80.6              | 42.5                    | 43.0             | 42.3 |
>
> MERCI discovers better, higher-quality solutions faster, utilizing training resources more effectively. In contrast, the baseline GRPO gets stuck in local optima, showing low exploration efficiency. When *vanilla GRPO* and *GRPO + our MERCI* is scaled to 260 steps, *GRPO+MERCI* exhibits a clear advantage in terms of Pass@k. Increasing Pass@k means the policy has successfully discovered and stabilized more new, successful reasoning paths that were previously hidden in the large search space. Finding these novel, high-reward paths is the definition of effective exploration.
>
> Furthermore, we are fully committed to providing the most comprehensive evidence; please let us know if there are any other specific quantitative metrics you would find particularly valuable for evaluating our algorithm.

---

> ### Author Response · Authors · 2025-11-21
> **Response to Reviewer 8z7R (3)**
>
> ## W3: Novelty and Distinction from Prior Work (e.g., Gao et al. [1])
> **We add the iMentor method (Gao et al. [1]) as a baseline in our new PDF, and the experimental results are shown in Table 1 and Table 2.** Our work is distinct from Gao et al. [1] and similar previous papers, primarily due to its theoretical foundation.
>
> 1.	A Novel Theoretical Framework (Contribution 1): Our main contribution is a new theoretical framework for LLM exploration. We are the first to formally show that because the LLM's transition dynamics are known and deterministic, the general Uncertainty Bellman Equation (UBE) dramatically simplifies. This simplification (Proposition 1) reduces the intractable problem of estimating Q-value uncertainty to the much more manageable problem of estimating local reward uncertainty ($V_t[\hat{r}^h(s)]$). This theoretical insight, to our knowledge, is novel.
> 2.	A Principled Method (MERCI): You correctly noted that many prior works, including some curiosity-driven methods like RND or ICM, share the high-level goal of exploration. As we state in our Introduction, many such curiosity-driven approaches lack a strong theoretical foundation on how the exploration bonus should decay. Our MERCI algorithm is a direct operationalization of this theoretical framework. We use a method (CFN) to estimate the pseudo-count because it is a theoretically sound proxy for the exact quantity our framework identifies, as $V_t[\hat{r}^h(s)] \propto 1/\mathcal{N}(s)$. This provides a principled, theoretically-grounded foundation for our method, distinguishing it from other (often heuristic) intrinsic motivation approaches like RND or ICM, or other works like [1] that share the high-level goal but lack this specific UBE-based derivation for LLMs.
> 3.	A Correct Bonus Calculation: Our derivation also leads to a theoretically correct method for bonus calculation: summing the variances and then taking the square root ($\sqrt{\sum \sigma^2}$). We explicitly contrast this with the "common but theoretically flawed heuristic" of summing standard deviations ($\sum \sqrt{\sigma^2}$), and we prove the superiority of our principled approach in our ablation study (Table 9 in Appendix G.2.4).
>
> In summary, our work provides the core theoretical justification (a simplified UBE for LLMs) and a principled, non-heuristic algorithm (MERCI) derived from it, constituting a significant novelty and contribution. We also further highlight our novelty in Section 4.2.
>
> We sincerely hope these clarifications adequately address your concerns and encourage you to reconsider your assessment of our work.

---

> ### Author Response · Authors · 2025-11-26
>
> We are very grateful for your thoughtful and detailed feedback on our manuscript. We believe that incorporating your suggestions has substantially improved the quality and clarity of the manuscript.
>
> As the discussion period is drawing to a close, we would like to confirm whether our responses have sufficiently addressed your concerns. Please do not hesitate to let us know if there are any further questions or areas for clarification. We are ready and willing to discuss any remaining concerns. We thank you once more for your invaluable contribution to our work.

---

> > ### Comment · Reviewer_8z7R · 2025-11-27
> >
> > Thanks the authors for addressing my concerns. I decide to increase the score.

---

> > > ### Author Response · Authors · 2025-11-28
> > >
> > > We are truly grateful for your engagement and for recognizing our work's contributions (raising the score from 2 to 6). Thank you once again for your valuable time and effort!

---

### Official Review · Reviewer_ZHHS · 2025-11-03

**Soundness:** 3
**Presentation:** 3
**Contribution:** 3
**Rating:** 8
**Confidence:** 4

**Summary:**

This paper introduces MERCI, a framework for motivating exploration in LLM reasoning using count-based intrinsic rewards. The method is grounded on a theoretical simplification of the Uncertainty Bellman Equation (UBE), which allows for estimation of trajectory uncertainty using a Coin Flip Network (CFN). The authors then convert the uncertainty into an intrinsic reward that can be integrated into RL algorithms like GRPO. Empirical evaluations on mathematical reasoning and SQL generation validate the effectiveness of MERCI.

**Strengths:**

1. Principled Theoretical Simplification: The paper provides a thoughtful application of the UBE to the LLM setting by removing the need to estimate environmental uncertainty. The reduction of Q-value uncertainty estimation to local reward uncertainty is precise and well-motivated.

2. The design of MERCI to use CFN for state visitation estimation is both conceptually clear and interesting.

3. The comparative results against GRPO, DAPO and the visualization of token-level estimated uncertainty support the claim proposed by authors.

**Weaknesses:**

- The bonus is scaled by a cosine schedule ($ \gamma $ decays to 10 % by step 200). There is no ablation on slower/fast decay, and no discussion of how this impacts the final performance.

- It seems that different reasoning tasks need to train the corresponding CFN (*e.g.*, MATH and SQL), which may limit its generalization ability. What is the computational cost of training a CFN? Can a single CFN be trained for tasks across different domains?

- I suggest citing relevant works [1][2][3] that also introduce intrinsic/curiosity reward bonuses to promote exploration.

---

[1] CDE: Curiosity-Driven Exploration for Efficient Reinforcement Learning in Large Language Models. arXiv preprint:2509.09675

[2] Consistent Paths Lead to Truth: Self-Rewarding Reinforcement Learning for LLM Reasoning. NeurIPS 2025

[3] Exploring Beyond Curiosity Rewards: Language-Driven Exploration in RL. PMLR 2025

**Questions:**

1. Can MERCI generalize to non-mathematical tasks such as MMLU-Pro, GPQA?

2. CFN may underestimate uncertainty for states that are linguistically close but not identical. Did the author observe or analyze any instances of failure?

3. Figure 8 shows the response length shrinks when MERCI is used. Is the policy truly “exploring”, or is the bonus merely penalising verbosity? I think an entropy-bonus scatter plot would help.

---

> ### Author Response · Authors · 2025-11-21
> **Response to Reviewer ZHHS (1)**
>
> We sincerely thank you for your time and insightful review on our submission.
>
> ## W1: Ablation on Bonus Decay Schedule
> We have conducted the requested ablation study, comparing our chosen Cosine decay schedule against faster and slower decay rates. The results are as follows:
>
> **pass@k:**
> | Model                     | AIME25 (pass@256) | AIME24 (pass@256) | Minerva (pass@16) | MATH500 (pass@16) | OlympiadBench (pass@16) | College (pass@8) | Avg. |
> | ------------------------- | ----------------- | ----------------- | ----------------- | ----------------- | ----------------------- | ---------------- | ---- |
> | GRPO + MERCI-decaystep200 | 60.0              | 80.0              | 63.2              | 91.4              | 60.9                    | 48.9             | 67.4 |
> | GRPO + MERCI-decaystep100 | 56.7              | 80.0              | 62.1              | 90.6              | 61.3                    | 48.6             | 66.6 |
> | GRPO + MERCI-decaystep300 | 56.7              | 76.7              | 62.9              | 90.8              | 60.3                    | 48.7             | 66.0 |
>
> **mean@k:**
> | Model                     | AIME25 (mean@256) | AIME24 (mean@256) | Minerva (mean@16) | MATH500 (mean@16) | OlympiadBench (mean@16) | College (mean@8) | Avg. |
> | ------------------------- | ----------------- | ----------------- | ----------------- | ----------------- | ----------------------- | ---------------- | ---- |
> | GRPO + MERCI-decaystep200 | 13.4              | 29.6              | 44.1              | 80.7              | 42.6                    | 42.9             | 42.2 |
> | GRPO + MERCI-decaystep100 | 12.9              | 28.9              | 43.8              | 80.0              | 43.1                    | 43.4             | 42.0 |
> | GRPO + MERCI-decaystep300 | 14.4              | 27.0              | 41.3              | 80.6              | 41.5                    | 42.4             | 41.2 |
>
> They are also presented in Table 10 in Appendix G.2.5. Our findings demonstrate that:
> 1. Optimal Balance: The Cosine decay of $\gamma$ to 10% by step 200 provides better overall performance.
> 2. Faster Decay: Schedules with a faster decay rate (e.g., decaying to 10% by step 100) led to insufficient exploration time. The policy quickly stabilized into suboptimal reasoning paths, resulting in a lower ceiling for the final performance.
> 3. Slower Decay: Conversely, schedules with a slower decay (e.g., decaying by step 300) hindered convergence late in training. The persistent, strong intrinsic reward introduced excessive noise or bias, preventing the policy from focusing on maximizing the external task reward, thus degrading the final performance and stability.
>
>
> ## W2: Generalization Ability and Training Cost of CFN
> The lightweight nature of the CFN, which attaches an MLP head on the generalized hidden-layer representations of the backbone LLM, makes it inherently flexible.
>
> We directly apply the CFN trained on mathematical reasoning tasks to estimate the uncertainty of responses in the SQL Generation task, and some examples are shown in Figure 8, 9, 10 (Appendix G.1) in our new PDF. Since SQL code is indeed infrequently encountered, it exhibits higher uncertainty, and the CFN correspondingly produces noticeably elevated estimates. In contrast, the uncertainty values assigned to other natural language reasoning sequences are largely consistent with our intuition and expectations.
>
> This directly demonstrates that the CFN is capable of leveraging the LLM's general features and translating them into an effective novelty estimate across domains. The pre-training of task-specific CFN in our paper was thus an optimization choice to maximize the in-task exploration performance, not a necessity imposed by limited generalization.
>
> As for the computational cost of training a CFN, the entire pre-training process requires approximately 30 minutes using 32 NVIDIA H20 GPUs. During training with policy model, it takes approximately 25% additional time using MERCI compared with the baseline methods.
>
>
> ## W3: Additional Citations
> Thank you for your suggestions. We have cited these relevant works in our new PDF (Section 4.2).

---

> ### Author Response · Authors · 2025-11-21
> **Response to Reviewer ZHHS (2)**
>
> ## Q1: Generalization to Non-mathematical Tasks
> Since we have not yet identified suitable RL training datasets tailored to the MMLU-Pro and GPQA domains, we directly evaluate the models trained on the Mathematical Reasoning dataset on MMLU-Pro and GPQA. The experimental results are as follows:
>
> | Model            | GPQA mean@16 | GPQA pass@8 | GPQA pass@16 | MMLU-Pro pass@1 |
> | ---------------- | ------------ | ----------- | ------------ | --------------- |
> | Qwen2.5-Math     | 8.98         | 47.5        | 53.6         | 5.8             |
> | + GRPO           | 24.3         | 59.8        | 61.6         | 28.3            |
> | + GRPO w / MERCI | 26.4         | 65.0        | 69.6         | 29.1            |
> | + DAPO           | 26.2         | 61.2        | 70.5         | 37.4            |
> | + DAPO w / MERCI | 27.4         | 64.3        | 73.7         | 39.5            |
>
>
> They are also included in Appendix G.2.1 in our new PDF. The results show that incorporating MERCI consistently provides gains on top of both GRPO and DAPO, particularly on GPQA pass@8 and pass@16, as well as MMLU-Pro pass@1. These findings suggest that our MERCI play an important role in improving out-of-domain robustness, even when the underlying training data is highly domain-specialized.
>
>
> ## Q2: Examples that are Linguistically Close but not Identical
> Some examples of reasoning trajectory and its corresponding uncertainty estimation are as follows (The uncertainty estimates are obtained from the CFN evaluations at step 80):
>
> 1. "1. **Total Number of Subsets:**\n   The set $S$ has $2^6 = 64$ subsets (since $S$ has 6 elements).\n\n2. **Choosing Subsets $A$ and $B$:**\n   Both $A$ and $B$ can be any subset of $S$. Therefore, there are $64 \times 64 = 4096$ possible pairs of $(A, B)$.\n\n3. **Condition for $B$ to be Contained in $A$ or $S-A$:**\n   For a given subset $A$, $B$ can be any subset of $A$ or any subset of $S-A$. The number of subsets of $A$ is $2^{|A|}$, and the number of subsets of $S-A$ is $2^{6-|A|}$. Therefore, for each subset $A$, there are $2^{|A|} + 2^{6-|A|} - 1$ valid subsets $B$": **0.2547**.
> 2. "1. **Counting Subsets:**\n   Since $S$ has 6 elements, its power set contains $2^6 = 64$ subsets.\n\n2. **Selecting $A$ and $B$:**\n   Both subsets are chosen uniformly from all 64 subsets, giving a total of $64 \times 64 = 4096$ ordered pairs $(A, B)$.\n\n3. **Requirement for $B$:**\n   Fix a particular $A$, $B$ is acceptable precisely when it is either a subset of $A$ or a subset of the complement $S-A$. There are $2^{|A|}$ subsets of $A$ and $S-A$ is $2^{6-|A|}$ subsets of $S-A$. Thus, the number of possible $B$'s is $2^{|A|} + 2^{6-|A|} - 1$": **0.2323**.
> 3. "1. **Elementwise Perspective:**\n   Instead of counting subsets directly, view the process per element: each element of $S$ independently lands in $A$, $B$, both, or neither.\n\n2. **Characterizing the Event:**\n   No element can simultaneously lie in $B$ and also in both $A$ and $S-A$. Thus, every element must choose from a restricted set of valid membership patterns.\n\n3. **Configuration Counting:**\n   For each element, determine how many assignments of membership in $A$ and $B$ satisfy the constraint; then raise that number to the 6th power": **0.3452**.
>
> The underlying rationale and semantics of (1) and (2) are linguistically close but not identical, and the resulting overall uncertainty estimates for the trajectories are also close. In contrast, (1) and (3) differ more substantially, leading to larger discrepancies in the aggregated trajectory-level uncertainty estimates. We also added these examples in Appendix G.1 in our new PDF.
>
>
> ## Q3: Entropy-Bonus Scatter Plot
> The entropy-bonus scatter plot is shown in Figure 12(a) in our new PDF. It indicates that positions at which CFN predicts higher uncertainty do not necessarily correspond to higher entropy. Policy entropy measures the policy's action randomness, whereas the CFN is designed to measure the model's epistemic uncertainty regarding its own state knowledge. Our results confirm that the CFN provides a unique, non-redundant signal that cannot be simply replaced by the policy's action probabilities.
>
> The response length shrinks probably because CFN penalizes low-novelty steps, which may be related to redundancy, and the length-normalization operation is applied. We posit that this may be attributable to optimizing for cognitive efficiency, which is a prerequisite for successful exploration in complex search spaces.

---

> > ### Comment · Reviewer_ZHHS · 2025-11-26
> >
> > Thanks for the authors' reply! I will keep my positive score.

---

> ### Author Response · Authors · 2025-11-26
>
> We are very grateful for your thoughtful and detailed feedback on our manuscript. We believe that incorporating your suggestions has substantially improved the quality and clarity of the manuscript.
>
> As the discussion period is drawing to a close, please do not hesitate to let us know if there are any further questions or areas for clarification. We are ready and willing to discuss any remaining concerns. We thank you once more for your invaluable contribution to our work.

---

> ### Author Response · Authors · 2025-11-26
>
> We are truly grateful for your engagement and for recognizing our work's contributions (keeping the positive score). Thank you once again for your valuable time and effort!

---

### Author Response · Authors · 2025-12-01
**Summary of Our Rebuttal and Reviewers' Feedback**

Dear Area Chair,

Thank you for the time and effort you have devoted to evaluating our submission, the reviews, and our rebuttal.

To help reduce your workload, we provide a brief summary of the key points from the reviewers as well as how our rebuttal addresses their concerns and how they reply.

# Summary of Reviewers' Comments and Our Rebuttal
Across all four reviews, the strengths consistently noted include the clear motivation, principled UBE-based formulation, lightweight CFN design, and steady empirical gains on math and SQL tasks.

At the same time, they raised some concerns.

**Reviewer ZHHS** noted missing ablations on the bonus-decay schedule and questioned CFN’s generalization, reliability on similar states, and computational cost. **We added corresponding experimental results in the rebuttal and manuscript.** Besides, reviewer ZHHS suggested some citations to related intrinsic-reward works, and we also cited them in our new manuscript.

**Reviewer 8z7R** might have a critical misunderstanding of our core method: *"The Coin Flipping Network Head is never trained"*. However, **our CFN is trained jointly with the policy network during the RL training process**, and in the rebuttal **we explained the principle of this joint training in detail**. In this regard, we realized that certain expressions in the manuscript might not be emphasized clearly enough, which could lead to misunderstandings for readers, so **we have highlighted this point in the manuscript**. In addition, regarding the additional experimental analyses that reviewer 8z7R requested and the further elaboration on the novelty, we also **provided detailed responses in the comment**.

**Reviewer XirT** requested experiments on the strict sum-then-sqrt setting (without group standardization) as well as experiments with different parameter choices. Reviewer XirT also questioned CFN’s state definition, CFN training overhead, handling of near-duplicates, and long-training stability. For the conceptual questions, we **provided more detailed and clearer explanations**; and for the **experimental results and analyses** that needed to be supplemented, we **added them both in the comment and in the new manuscript**.

**Reviewer T5WN** pointed out missing comparisons with recent exploration baselines, and requested deeper analysis of our MERCI’s effect on reasoning behaviors. To address these issues, we **added comparative experiments with the two baselines** mentioned by the reviewer on **both Math and SQL datasets**, and verified that our method **performs better**. In addition, we also provided a more detailed case study to illustrate that **our MERCI contributes to finding more elegant and insightful solutions**.



# Summary of Reviewers' Feedback on Our Rebuttal

Before November 27th, we have received responses from reviewer ZHHS (initial score: 8), reviewer XirT (initial score: 6) and reviewer T5WN (initial score: 4), all of whom indicated that their concerns had been adequately addressed. Shortly after the bug occurred, reviewer 8z7R (initial score: 2) also responded to our rebuttal, likewise acknowledging that concerns had been resolved.

**Reviewer XirT** first acknowledged our response and affirmed the results of our additional experiments, and **raised the score from 6 to 8 on November 23rd** (https://openreview.net/revisions?id=aOcUyirRN8). **On November 26th**, after we further invited the reviewers to participate, **reviewer ZHHS** responded that same day, acknowledged our rebuttal, and **maintained the score of 8**. **Reviewer T5WN** also stated that we had addressed concerns and **raised the score from 4 to 6 on November 26th** (https://openreview.net/revisions?id=iAbeP2B16w).

In the initial review, **reviewer 8z7R had a critical misunderstanding regarding the design of our method**, which directly resulted in questions on the effectiveness and novelty of our approach. In the rebuttal, we provided a very detailed explanation and presented more experimental results，and emphasized this point in the manuscript to avoid possible misunderstandings. Although **reviewer 8z7R increased score from 2 to 6** after the bug (https://openreview.net/revisions?id=2kWaGOKSOT), we believe this was due to our further clarification of the method and the additional experiments that addressed reviewer 8z7R's concerns.

In summary, **before the bug occurred, our average score had already been increased to 6 (2,6,8,8)**; and **before the scores were reverted, our average score had already been increased to 7 (6,6,8,8)**.

For the specific details, you may refer to our **discussion record**.

Once again, we sincerely appreciate the additional time and effort the Area Chair have put in evaluating our submission. We are also deeply grateful to all reviewers for their recognition of our work and for the constructive and insightful suggestions that have helped us improve our manuscript.

Best Regards,

ICLR 2026 Conference Submission24350 Authors

---

> ### Author Response · Authors · 2025-12-02
> **Summary of Our Work's Strengths**
>
> Dear Area Chair,
>
> Thank you for the time and effort you have devoted to evaluating our submission, the reviews, and our rebuttal.
>
> In addition to summarizing our rebuttal and reviewers' feedback, we also provide a brief summary of our work's strengths to help you quickly understand our contributions.
>
> # Summary of Our Work's Strengths
>
> The reviewers had already highlighted many strengths in their original reviews, and after we further improved the manuscript based on their comments, our work now possesses very comprehensive strengths:
>
> * **Clear motivation and writing**: Reviewers consistently praise our manuscript for being well-motivated, clearly presented, and easy to follow.
>
> * **Addresses an important problem**: Our work tackles the exploration challenge in LLM reasoning and RLVR, which reviewers describe as timely, significant, and underexplored.
>
> * **Principled theoretical foundation**: The simplification of the Uncertainty Bellman Equation under deterministic transitions is seen as insightful and provides a solid basis for our intrinsic reward design.
>
> * **Lightweight and elegant CFN design**: Reviewers appreciate that the Coin Flipping Network offers an efficient and simple mechanism for estimating pseudo-counts (epistemic uncertainty).
>
> * **Empirical validation of CFN reliability**: Results in our rebuttal show strong correlation between CFN bonuses and actual uncertainty, and the CFN can also distinguish near-duplicate states and generalize to unseen domains. This strengthens the credibility of the CFN mechanism.
>
> * **Broad and consistent empirical gains**: Our algorithm shows steady improvements across multiple mathematical-reasoning and SQL benchmarks, and also demonstrates cross-domain applicability. Additionally, our method performs better than recent exploration baselines.
>
> * **Compatibility with existing RL methods**: MERCI is validated on both GRPO and DAPO, which reviewers view as evidence of its flexibility and general usefulness.
>
> * **Useful visual and empirical analyses**: Examples, rollout visualizations, and training-dynamics plots were noted as helpful in illustrating MERCI's behavior and impact. Additional analyses included in the rebuttal more clearly show how our MERCI contributes to performing high-level reasoning patterns and finding more elegant and insightful solutions.
>
> Overall, our work **offers both conceptual and empirical contributions to exploration in LLM-based RL**. We hope that the clarified contributions and our polished manuscript meaningfully demonstrate the value of our work.
>
> Best Regards,
>
> ICLR 2026 Conference Submission24350 Authors

---

### Meta-Review · Area_Chair_wv7q · 2026-01-02

**Summary:**

This paper addresses the challenge of exploration RL for LLM reasoning. Prevalent RL paradigms rely on sparse, outcome-based rewards and limited exploration. The authors introduce MERCI to augment policy optimization with a principled intrinsic reward. Building on the idea of count-based exploration, MERCI leverages a lightweight Coin Flipping Network (CFN) to estimate pseudo-counts and epistemic uncertainty over reasoning trajectories, converting these into an intrinsic reward that values novelty while preserving the learning signal from task rewards.

**Reviewer Concerns:**

1. It would be fair to also show MERCI‑scaling beyond 120 steps to check stability under longer training.
2. Missing baselines: There have been many works trying to solve the exploration problem, such as i-MENTOR [1] and entropy-based advantage shaping [2].

**Reviewer Scores:**

The paper received mixed ratings before the rebuttal. The reviewers think the idea is novel and the results are promissing. However, there are some mis-alignment and concerns from the reviewers. The rebuattl has addressed all the concerns and mist-understanding. The final decision is acceptance.

---

### Decision · Program_Chairs · 2026-01-26

Accept (Poster)